# Object Detection with Transformers: A Review

**DOI:** 10.3390/s25196025

**Published:** 2025-10-01

**Authors:** Tahira Shehzadi, Khurram Azeem Hashmi, Marcus Liwicki, Didier Stricker, Muhammad Zeshan Afzal

**Affiliations:** 1Department of Computer Science, Technical University of Kaiserslautern, 67663 Kaiserslautern, Germanymuhammad_zeshan.afzal@dfki.de (M.Z.A.); 2Mindgarage Lab, Technical University of Kaiserslautern, 67663 Kaiserslautern, Germany; 3German Research Institute for Artificial Intelligence (DFKI), 67663 Kaiserslautern, Germany; 4Department of Computer Science, Electrical and Space Engineering, Luleå University of Technology, 971 87 Luleå, Sweden

**Keywords:** transformer, object detection, DETR, computer vision, deep neural networks

## Abstract

The astounding performance of transformers in natural language processing (NLP) has motivated researchers to explore their applications in computer vision tasks. A detection transformer (DETR) introduces transformers to object detection tasks by reframing detection as a set prediction problem. Consequently, it eliminates the need for proposal generation and post-processing steps. Despite competitive performance, DETR initially suffered from slow convergence and poor detection of small objects. However, numerous improvements are proposed to address these issues, leading to substantial improvements, enabling DETR to achieve state-of-the-art performance. To the best of our knowledge, this paper is the first to provide a comprehensive review of 25 recent DETR advancements. We dive into both the foundational modules of DETR and its recent enhancements, such as modifications to the backbone structure, query design strategies, and refinements to attention mechanisms. Moreover, we conduct a comparative analysis across various detection transformers, evaluating their performance and network architectures. We aim for this study to encourage further research in addressing the existing challenges and exploring the application of transformers in the object detection domain.

## 1. Introduction

Object detection is a fundamental task in computer vision that involves locating and classifying objects within an image [1,2,3,4,5,6], with applications in autonomous driving, surveillance, robotics, and medical imaging. In autonomous driving, for example, accurately detecting pedestrians, vehicles, and traffic signs in real time is critical for safety. Traditionally, convolutional neural networks (CNNs), such as faster R-CNN [1] and RetinaNet [4], have served as the primary backbones for object detection models, achieving impressive performance. However, these models heavily rely on hand-crafted components like region proposal networks (RPNs) and post-processing steps such as non-maximum suppression (NMS) [7], which complicate the training pipeline and limit end-to-end optimization. The recent success of transformers in natural language processing (NLP) has motivated researchers to explore their potential in computer vision [8]. The transformer architecture [9,10] effectively captures long-range dependencies in sequential data, enabling global context modeling that is difficult for traditional CNNs. This capability makes transformers particularly attractive for object detection, where recognizing objects often depends on global context.

The transformer architecture [9,10] is characterized by its encoder–decoder structure and the use of self-attention and cross-attention mechanisms, which allow it to capture long-range dependencies across input sequences effectively. Unlike CNNs, which primarily focus on local features through convolutional kernels, transformers can model global relationships across an entire image. This capability makes transformers particularly suitable for object detection, where understanding the spatial and contextual relationships between multiple objects is crucial. Leveraging this strength, researchers have explored transformer-based approaches to develop end-to-end object detection frameworks that do not rely on hand-crafted components.

In this context, Carion et al. (2020) proposed the detection transformer (DETR) [11], a novel framework that replaces traditional region proposal-based methods with a end-to-end trainable architecture using a transformer encoder–decoder network. The DETR network demonstrates promising performance, outperforming conventional CNN-based object detectors [12,13,14,15,16,17,18,19], while also eliminating the need for components such as region proposal networks and post-processing steps like non-maximum suppression (NMS) [7]. Despite these advantages, DETR has certain limitations, including slow training convergence and reduced performance on small objects, which have motivated numerous modifications and improvements in subsequent research.

Since DETR’s introduction, numerous variants have emerged to address limitations such as slow convergence, small object detection, and computational efficiency. Figure 1 illustrates the growth and evolution of DETR research, showing rising publications and citations, widespread architectural modifications, and a focus on key challenges like improving training stability, efficiency, and small object performance. This highlights the rapid expansion of transformer-based detection, emphasizing the need for a comprehensive review, to which numerous DETR variants have responded. Deformable-DETR [20] modifies the attention modules to process the image feature maps by considering the attention mechanism as the main reason for slow training convergence, while UP-DETR [21] proposes modifications to pre-train DETR similar to the pre-training of transformers in natural language processing. Efficient-DETR [22], based on original DETR and Deformable-DETR, examines the randomly initialized object probabilities, including reference points and object queries, which is one of the reasons for multiple training iterations. SMCA-DETR [23] introduces a spatially modulated co-attention module that replaces the existing co-attention mechanism in DETR to overcome slow training convergence, and TSP-DETR [24] deals with cross-attention and the instability of bipartite matching. Conditional-DETR [25] presents a conditional cross-attention mechanism, while WB-DETR [26] considers a CNN backbone for feature extraction as an extra component and presents a transformer encoder–decoder network without a backbone. PnP-DETR [27] proposes a PnP sampling module to reduce spatial redundancy and improve computational efficiency. Dynamic-DETR [28] introduces dynamic attention in the encoder–decoder network, YOLOS-DETR [29] demonstrates the transferability and versatility of the transformer from image recognition to detection, Anchor-DETR [30] proposes object queries as anchor points, Sparse-DETR [31] reduces computational cost via token filtering, D^2^ETR [32] uses cross-scale attention in the decoder, FP-DETR [33] reformulates pre-training and fine-tuning, and CF-DETR [34] refines predicted locations to improve small object detection. Further improvements targeting training stability and small object performance include DN-DETR [35], which uses noised object queries as additional decoder input to reduce the instability of the bipartite-matching mechanism, AdaMixer [36], which considers the encoder an extra network between the backbone and decoder and introduces a 3D sampling process, REGO-DETR [37], which proposes an RoI-based method for detection refinement, and DINO [38], which uses positive and negative noised object queries to accelerate convergence and enhance performance on small objects. These successive innovations collectively address the limitations of the original DETR while retaining its advantages as a fully end-to-end transformer-based object detector. FP-DETR [33] reformulates the pre-training and fine-tuning stages for detection transformers. CF-DETR [34] refines the predicted locations by utilizing local information, as incorrect bounding box location reduces performance on small objects.

DN-DETR [35] uses noised object queries as additional decoder input to reduce the instability of the bipartite-matching mechanism in DETR, which causes the slow convergence problem. AdaMixer [36] considers the encoder an extra network between the backbone and decoder that limits the performance and slows the training convergence because of its design complexity. It proposes a 3D sampling process and a few other modifications in the decoder. REGO-DETR [37] proposes an RoI-based method for detection refinement to improve the attention mechanism in the detection transformer. DINO [38] considers positive and negative noised object queries to make training convergence faster and to enhance the performance on small objects. Building on these improvements, Co-DETR [39] introduces collaborative hybrid assignments to improve training stability and convergence speed, addressing limitations in bipartite matching and small object performance. LW-DETR [40] focuses on efficiency, using a lightweight ViT encoder, a shallow decoder, and global attention to reduce computational cost while maintaining competitive accuracy. RT-DETR [41] combines a hybrid encoder with multi-scale feature processing and IoU-aware query selection to achieve adaptable inference speed, balancing high accuracy with real-time performance.

The rapid pace of advancements makes it difficult to track progress systematically. Thus, a review of ongoing progress is necessary and would be helpful for the researchers in the field. This paper provides a detailed overview of recent advancements in detection transformers. Table 1 shows the overview of Detection Transformer (DETR) modifications to improve performance and training convergence. Many surveys have studied deep learning approaches in object detection [42,43,44,45,46,47]. Table 2 lists existing object detection surveys. Among these, several studies comprehensively review approaches that process different 2D data types [48,49,50,51], while others focus on specific 2D applications [52,53,54,55,56,57,58,59] or related tasks such as segmentation [60,61,62], image captioning [63,64,65,66], and object tracking [67]. Furthermore, some surveys examine deep learning methods and introduce vision transformers [68,69,70,71]. Nonetheless, most of these surveys were published before recent improvements in detection transformer networks, and a comprehensive review of transformer-based object detectors is still lacking. Therefore, a detailed survey of ongoing advancements is necessary to provide guidance and insights for researchers.

A detailedreview of transformer-based detection methods from an architectural perspective. We categorize and summarize improvements in the detection transformer (DETR) according to backbone modifications, pre-training level, attention mechanism, query design, etc. This analysis aims to help researchers develop a deeper understanding of the key components of detection transformers in terms of performance indicators.A performance evaluation of detection transformers. We evaluate improvements in detection transformers using the popular benchmark MS COCO [75]. We also highlight the advantages and limitations of these approaches.An analysis of accuracy and computational complexity of improved versions of detection transformers. We present an evaluative comparison of state-of-the-art transformer-based detection methods with respect to attention mechanisms, backbone modifications, and query designs.An overview of the key building blocks of detection transformers to improve their performance further and future directions. We examine the impact of various key architectural design modules that impact network performance and training convergence to provide possible suggestions for future research. Readers interested in ongoing developments in detection transformers can refer to our Github repository; https://github.com/mindgarage-shan/transformer_object_detection_survey (accessed on 25 September 2025).

The remaining paper is arranged as follows. Section 2 is related to object detection and transformers in all types of vision. Section 3 is the main part, which explains the modifications in the detection transformers in detail. Section 3.24 refers to the evaluation protocol, and Section 4 provides a comparative evaluation of detection transformers. Section 5 discusses open challenges and future directions. Finally, Section 6 concludes the paper.

## 2. Object Detection and Transformers in Vision

### 2.1. Object Detection

This section explains the key concept of object detection and previously used object detectors. A more detailed analysis of object detection concepts can be found in [74,76,77]. The object detection task localizes and recognizes objects in an image by providing a bounding box around each object and its category. These detectors are usually trained on datasets like PASCAL VOC [78] or MS COCO [75]. The backbone network extracts the features of the input image as feature maps [79]. Usually, the backbone network, such as ResNet-50 [80], is pre-trained on ImageNet [81] and then fine-tuned to downstream tasks [82,83,84,85,86,87]. Moreover, many works have also used visual transformers [3,88,89] as a backbone. Single-stage object detectors [3,4,90,91,92,93,94,95,96,97,98] use only one network, having faster speed but lower performance than two-stage networks. Two-stage object detectors [1,2,7,79,99,100,101,102,103,104] contain two networks, which provide final bounding boxes and class labels.

**Lightweight Detectors:** Lightweight detectors are designed to be more computationally efficient than standard object detection models. These are real-time object detectors and can be employed on small devices. Examples include [105,106,107,108,109,110,111,112,113,114].

**Three-Dimensional Object Detection:** The primary purpose of 3D object detection is to recognize the objects of interest using a 3D bounding box and give a class label. Three-dimensional approaches fall into three categories: image-based [115,116,117,118,119,120,121], point cloud-based [122,123,124,125,126,127,128,129,130], and multimodal fusion-based [131,132,133,134,135].

### 2.2. Transformer for Segmentation

The self-attention mechanism can be employed for segmentation tasks [136,137,138,139,140] that provide pixel-level [141] prediction results. Panoptic segmentation [142] jointly solves semantic and instance segmentation tasks by providing per-pixel class and instance labels. Wang et al. [143] propose location-sensitive axial attention for the panoptic segmentation task on three benchmarks [75,144,145]. The above segmentation approaches have self-attention in CNN-based networks. Recently, segmentation transformers [137,139] containing encoder– decoder modules have provided new directions to employ transformers for segmentation tasks.

### 2.3. Transformers for Scene and Image Generation

Previously, text-to-image generation methods [146,147,148,149] were based on GANs [150]. Ramesh et al. [151] introduced a transformer-based model for generating high-quality images from provided text details. Transformer networks are also applied for image synthesis [152,153,154,155,156], which is important for learning unsupervised and generative models for downstream tasks. Feature learning with an unsupervised training procedure [153] achieves state-of-the-art performance on two datasets [157,158], while SimCLR [159] provides comparable performance on [160]. The iGPT image generation network [153] does not include pre-training procedures similar to language modeling tasks. However, unsupervised CNN-based networks [161,162,163] consider prior knowledge as the architectural layout, attention mechanism, and regularization. Generative adversarial networks (GAN) [150] with CNN-based backbones are appealing for image synthesis [164,165,166]. TransGAN [155] is a strong GAN network where the generator and discriminator contain transformer modules. These transformer-based networks boost performance for scene and image generation tasks.

### 2.4. Transformers for Low-Level Vision

Low-level vision analyzes images to identify their basic components and create an intermediate representation for further processing and higher-level tasks. After observing the remarkable performance of attention networks in high-level vision tasks [11,137], many attention-based approaches have been introduced for low-level vision problems, such as [167,168,169,170,171].

### 2.5. Transformers for Multi-Modal Tasks

Multi-modal tasks involve processing and combining information from multiple sources or modalities, such as text, images, audio, or video. The application of transformer networks in vision language tasks has also been widespread, including visual question-answering [172], visual commonsense-reasoning [173], cross-modal retrieval [174], and image captioning [175]. These transformer designs can be classified into single-stream [176,177,178,179,180,181] and dual-stream networks [182,183,184]. The primary distinction between these networks lies in the choice of loss functions.

## 3. Detection Transformers

This section briefly explains the detection transformer (DETR) and its improvements, as shown in Figure 2.

### 3.1. DETR

The detection transformer (DETR) [11] architecture is much simpler than CNN-based detectors like faster R-CNN [185] as it removes the need for an anchor generation process and post-processing steps, such as non-maximal suppression (NMS), and provides an optimal detection framework. The DETR network has three main modules: a backbone network with positional encodings, an encoder, and a decoder network with an attention mechanism. The extracted features from the backbone network are a single vector and its positional encoding [186,187] within the input vector fed to the encoder network. Here, self-attention is performed on key, query, and value matrices forwarded to the multi-head attention and feed-forward network to find the attention probabilities of the input vector. The DETR decoder takes object queries in parallel with the encoder output. It computes predictions by decoding N number of object queries in parallel. It uses a bipartite-matching algorithm to label the ground-truth and predicted objects, as provided in the following equation:(1)$$\hat{\sigma }=\arg\min_{\sigma \in N}\sum_{k}^{N}L_{m}\left(y_{k},{\hat{y}}_{\sigma (k)}\right).$$

Here, $y_{k}$ is a set of ground-truth (GT) objects. It provides boxes for both object and “no object” classes, where *N* is the total number of objects to be detected. $L_{m}\left(y_{k},{\hat{y}}_{\sigma (k)}\right)$ represents the duplicate-free matching cost between predicted objects σ(k) and ground-truth $y_{k}$, as defined below:(2)$$L_{m}\left(y_{k},{\hat{y}}_{\sigma (k)}\right)=-1_{\left\{c_{k}\neq \phi \right\}}{\hat{p}}_{\sigma (k)}\left(c_{k}\right)+1_{\left\{c_{k}\neq \phi \right\}}L_{bbox}\left(b_{k},{\hat{b}}_{\hat{\sigma }}\left(k\right)\right).$$

The next step is to compute the Hungarian loss by determining the optimal matching between ground-truth (GT) and detected boxes regarding the bounding-box region and label. The loss is reduced by stochastic gradient descent (SGD).(3)$$L_{H}\left(y,\hat{y}\right)=\sum_{k=1}^{N}\left[-log{\hat{p}}_{\hat{\sigma }\left(k\right)}\left(c_{k}\right)+1_{\left\{c_{k}\neq \phi \right\}}L_{box}\left(b_{k},{\hat{b}}_{\hat{\sigma }}\left(k\right)\right)\right],$$
where ${\hat{p}}_{\hat{\sigma }\left(k\right)}$ and $c_{k}$ are the predicted class and target label, respectively. The term $\hat{\sigma }$ is the optimal-assignment factor; $b_{k}$ and ${\hat{b}}_{\hat{\sigma }}\left(k\right)$ are ground-truth and predicted bounding boxes. The term $\hat{y}$ and $y=\{\left(c_{k},b_{k}\right)\}$ are the prediction and ground-truth of objects, respectively. Specifically, the bounding box loss is a linear combination of the generalized IoU (GIoU) loss [188] and of the L1 loss, as in the following equation:(4)$$L_{bbox}=\lambda_{i}L_{iou}\left(b_{k},{\hat{b}}_{\sigma (k)}\right)+\lambda_{l1}{\left\|b_{k}-{\hat{b}}_{\sigma (k)}\right\|}_{1},$$
where $\lambda_{i}$ and $\lambda_{l1}$ are the hyperparameters. DETR can only predict a fixed number of *N* objects in a single pass. For the COCO dataset [75], the value of *N* is set to 100 as this dataset has 80 classes. This network does not need NMS to remove redundant predictions as it uses bipartite matching loss with parallel decoding [189,190,191]. In comparison, previous studies used RNN-based autoregressive decoding [192,193,194,195,196,196]. The DETR network has several challenges, such as slow training convergence and performance drops for small objects. To address these challenges, modifications have been made to the DETR network. Despite its end-to-end design, DETR suffers from slow training convergence and lower accuracy for small objects. The uniform attention initialization and lack of multi-scale features make learning precise object locations difficult. These limitations motivated the development of several modifications aimed at improving convergence, computational efficiency, and small object detection.

### 3.2. Deformable-DETR

The attention module of DETR provides a uniform weight value to all pixels of the input feature map at the initialization stage. These weights need many epochs for training convergence to find informative pixel locations. However, it requires high computation and extensive memory. The encoder’s self-attention has complexity $O(w_{i}^{2}h_{i}^{2}c_{i})$. In contrast, the decoder’s cross-attention has complexity $O(h_{i}w_{i}c_{i}^{2}+Nh_{i}w_{i}c_{i})$. Formally, $h_{i}$ and $w_{i}$ denote the height and width of the input feature map, respectively, and N represents object queries fed as input to the decoder. Let $q\in \Omega_{q}$ denote a query element with feature $z_{q}\in R^{c_{i}}$, and $k\in \Omega_{k}$ represents a key vector with feature $x_{k}\in R^{c_{i}}$, where $c_{i}$ is the input features dimension, $\Omega_{k}$ and $\Omega_{q}$ indicate the set of key and query vectors, respectively. Then, the feature of multi-head attention (MHAttn) is computed by the following:(5)$$MHAttn\left(z_{q},x\right)=\sum_{j=1}^{J}W_{j}\left[\sum_{k\in \Omega_{k}}A_{jqk}.W_{j}^{\prime }x_{k}\right],$$
where *j* represents the attention head, $W_{j}\in R^{c_{i}\times c_{v}}$, and $W_{j}^{\prime }\in R^{c_{v}\times c_{i}}$ are of learnable weights ($c_{v}=c_{i}/J$ by default). The attention weights $A_{jqk}\propto exp\frac{z_{q}^{T}U_{j}^{T}V_{j}x_{k}}{{\sqrt{c}}_{v}}$ are normalized as $\sum_{k\in \Omega_{k}}A_{jqk}=1$, in which $U_{j},V_{j}\in R^{c_{v}\times c_{i}}$ are also learnable weights. Deformable-DETR [20] modifies the attention modules inspired by [197,198] to process the image feature map by considering the attention network as the main reason for slow training convergence and confined feature spatial resolution. This module samples a small set of features near each reference point. Given an input feature map $x\in R^{c_{i}\times h_{i}\times w_{i}}$, let query q have content feature $z_{q}$ and a 2D reference point $r_{q}$, and the deformable attention feature is computed by the following: (6)$$DeformAttn\left(z_{q},r_{q},x\right)=\sum_{j=1}^{J}W_{j}\left[\sum_{k=1}^{K}A_{jqk}.W_{j}x\left(r_{q}+\Delta r_{jqk}\right)\right],$$
where $\Delta r_{jqk}$ indexes the sampling offset. It takes ten times fewer training epochs than a simple DETR network. The complexity of self-attention becomes $O(w_{i}h_{i}c_{i}^{2})$, which is linear complexity according to spatial size $h_{i}w_{i}$. The complexity of the cross-attention in the decoder becomes $O(NKc_{i}^{2})$, which is independent of spatial size $h_{i}w_{i}$. In Figure 3, the dark pink block indicates the deformable attention module in Deformable-DETR.

**Multi-Scale Feature Maps:** High-resolution input image features increase the network efficiency, specifically for small objects. However, this is computationally expensive. Deformable-DETR provides high-resolution features without affecting the computation. It uses a feature pyramid containing high and low-resolution features rather than the original high-resolution input image feature map. This feature pyramid has an input image resolution of 1/8, 1/16, and 1/32 and contains its relative positional embeddings. Furthermore, Deformable-DETR replaces the attention module in DETR with the multi-scale deformable attention module to reduce computational complexity and improve performance. While Deformable-DETR accelerates training and improves small object detection, designing effective sampling offsets and managing multi-scale feature interactions remain critical to achieving optimal performance. Algorithm 1 illustrates the step-by-step implementation of the multi-scale deformable attention mechanism, complementing the mathematical formulation presented above.
**Algorithm 1:** Multi-scale deformable attention in Deformable-DETR.
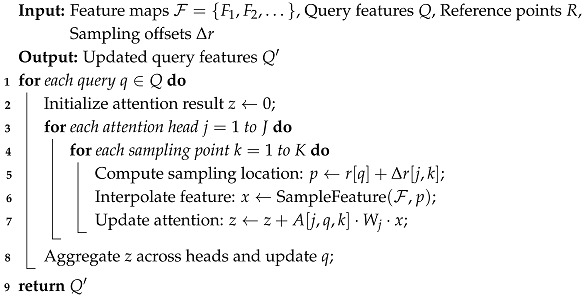


### 3.3. UP-DETR

Dai et al. [21] proposed a few modifications to pre-train the DETR similar to pre-training transformers in NLP. The randomly sized patches from the input image are used as object queries to the decoder as input. The pre-training proposed by UP-DETR helps to detect these randomly sized query patches. Algorithm 2 summarizes the pre-training procedure of UP-DETR, illustrating how random patches, query grouping, and attention masking are applied to improve convergence and feature learning. In Figure 3, the bright cyan block denotes UP-DETR. Two issues are addressed during pre-training: multi-task learning and multi-query localization.

**Multi-Task Learning:** The object detection task combines object localization and classification, while these tasks always have distinct features [199,200,201]. The patch detection damages the classification features. Multi-task learning using patch feature reconstruction and a frozen pre-training backbone is proposed to protect the classification features of the transformer. The feature reconstruction is given as follows:(7)$$L_{rec}\left(f_{k},{\hat{f}}_{\hat{\sigma }\left(k\right)}\right)={\left\|\frac{f_{k}}{\|f_{k}{\|}_{2}}-\frac{{\hat{f}}_{\hat{\sigma }\left(k\right)}}{\|{\hat{f}}_{\hat{\sigma }\left(k\right)}{\|}_{2}}\right\|}_{2}^{2}.$$

Here, the feature reconstruction term is $L_{rec}$. It is the mean-squared error between $l_{2}$ (normalized) features of patches obtained from the CNN backbone.

**Multi-query Localization:** The decoder of DETR takes object queries as input to focus on different positions and box sizes. When this object queries a number *N* (typically N=100) that is high, a single-query group is unsuitable as it has convergence issues. To solve the multi-query localization problem between object queries and patches, UP-DETR proposes an attention mask and query shuffle mechanism. The number of object queries is divided into *X* different groups, where each patch is provided to N/X object queries. The Softmax layer of the self-attention module in the decoder is modified by adding an attention mask inspired by [202] as follows:(8)$$P\left(q_{i},k_{i}\right)=Softmax\left(\frac{q_{i}k_{i}^{T}}{\sqrt{d}}+M\right)v_{i},$$(9)$$M_{k,l}=\left\{\begin{array}{cc} 0 & \;k,l\;in\;the\;same\;group\\ -\infty & \;otherwise \end{array}\right.,$$
where $M_{k,l}$ is the interaction parameter of object queries $q_{k}$ and $q_{l}$. Though object queries are divided into groups, these queries do not have explicit groups during downstream training tasks. Therefore, these queries are randomly shuffled during pre-training by masking 10% query patches to zero, similar to dropout [203]. Although UP-DETR improves convergence and query learning, the pre-training may not always transfer perfectly to downstream detection tasks, and its grouping and masking mechanisms require careful tuning to avoid convergence or performance issues. Algorithm 2 shows the patch detection pre-training procedure, where random patches are cropped, assigned to query subsets with attention masking, and the model is trained to predict patch locations while reconstructing features, improving robustness and convergence.
**Algorithm 2:** Patch detection pre-training in UP-DETR.
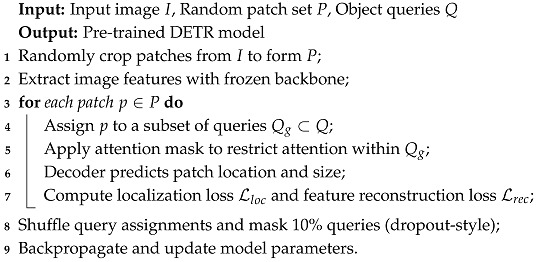


### 3.4. Efficient-DETR

The performance of DETR also depends on the object queries, as the detection head obtains final predictions from them. However, these object queries are randomly initialized at the start of training. Efficient-DETR [22], based on DETR and Deformable-DETR, examines the randomly initialized object blocks, including reference points and object queries, which is one of the reasons for multiple training iterations. In Figure 3, the dull green box shows Efficient-DETR.

Efficient-DETR has two main modules: a dense module and a sparse module. These modules have the same final detection head. The dense module includes the backbone network, encoder network, and detection head. Following [204], it generates proposals by a class-specific dense prediction using the sliding window and selects Top-k features as object queries and reference points. Efficient-DETR uses 4D boxes as reference points rather than 2D centers. The sparse network does the same work as the dense network, except for their output size. The features from the dense module are taken as the initial state of the sparse module, which is considered a good initialization of object queries. Both dense and sparse modules use a one-to-one assignment rule, as in [205,206,207]. However, Efficient-DETR adds architectural complexity, and the final performance heavily depends on the quality of the dense module’s proposals, making the approach sensitive to the selection of initial object queries and hyperparameters.

### 3.5. SMCA-DETR

The decoder of the DETR takes object queries as input that are responsible for object detection in various spatial locations. These object queries combine with spatial features from the encoder. The co-attention mechanism in DETR involves computing a set of attention maps between the object queries and the image features to provide class labels and bounding box locations. However, the visual regions in the decoder of DETR related to object query might be irrelevant to the predicted bounding boxes. This is one of the reasons that DETR needs many training epochs to find suitable visual locations to identify corresponding objects correctly. Gao et al. [23] introduced a spatially modulated co-attention (SMCA) module that replaces the existing co-attention mechanism in DETR to overcome the slow training convergence of DETR. In Figure 4, the purple block represents SMCA-DETR. The object queries estimate the scale and center of its corresponding object, which are further used to set up a 2D spatial weight map. The initial estimate of scale $l_{h_{i}},l_{w_{i}}$ and center $e_{h_{i}},e_{w_{i}}$ of a Gaussian-like distribution for object queries *q* is provided as follows:(10)$$\begin{array}{cc} e_{h_{i}}^{nrm},e_{w_{i}}^{nrm} & =sigmoid(MLP(q)), \end{array}$$(11)$$\begin{array}{cc} l_{h_{i}},l_{w_{i}} & =FC(q), \end{array}$$
where object query *q* provides a prediction center in normalized form by a sigmoid activation function after two layers of MLP. These predicted centers are un-normalized to obtain the input image’s center coordinates $e_{h_{i}}$ and $e_{w_{i}}$. The object query also estimates the object scales as $l_{h_{i}}$ and $l_{w_{i}}$. After the prediction of the object scale and center, SMCA provides a Gaussian-like weight map as follows:(12)$$W\left(x,y\right)=exp\left(-\frac{{\left(x-e_{w_{i}}\right)}^{2}}{\beta l_{w_{i}}^{2}}-\frac{{\left(y-e_{h_{i}}\right)}^{2}}{\beta l_{h_{i}}^{2}}\right),$$
where β is the hyperparameter to regulate the bandwidth, and (x,y) is the spatial parameter of weight map W. It provides high attention to spatial locations closer to the center and low attention to spatial locations away from the center.(13)$$A_{i}=Softmax\left(\frac{q_{i}k_{i}^{T}}{\sqrt{d}}+\log W\right)v_{i}.$$

Here, $A_{i}$ is the co-attention map. The difference between the co-attention module in DETR and this co-attention module is the addition of the logarithm of the spatial-map W. The decoder attention network has more attention near predicted box regions, which limits the search locations and thus converges the network faster. SMCA-DETR improves training efficiency and small object detection. However, its success depends on accurate initial predictions of object centers and scales, making it sensitive to initialization and hyperparameters.

### 3.6. TSP-DETR

TSP-DETR [24] deals with the cross-attention and the instability of bipartite matching to overcome the slow training convergence of DETR. TSP-DETR proposes two modules based on an encoder network with feature pyramid networks (FPN) [79] to accelerate the training convergence of DETR. In Figure 4, the orange block indicates TSP-DETR. These modules are TSP-FCOS and TSP-RCNN, which use a classical one-stage detector FCOS [208] and classical two-stage detector Faster-RCNN [1], respectively. TSP-FCOS used a new Feature of Interest (FoI) module to handle the multi-level features in the transformer encoder. Both modules use the bipartite matching mechanism to accelerate the training convergence.

**TSP-FCOS:** The TP-FCOS module follows the FCOS [208] for designing the backbone and FPN [79]. Firstly, the features extracted by the CNN backbone from the input image are fed to the FPN component to produce multi-level features. Two feature extraction heads, the classification head and the auxiliary head, use four convolutional layers and group normalization [209], which are shared across the feature pyramid stages. Then, the FoI classifier filters the concatenated output of these heads to select top-scored features. Finally, the transformer encoder network takes these FoIs and their positional encodings as input, providing class labels and bounding boxes as output.

**TSP-RCNN:** Like TP-FCOS, this module extracts the features from the CNN backbone and produces multi-level features using the FPN component. In place of two feature extraction heads used in TSP-FCOS, the TSP-RCNN module follows the design of faster R-CNN [1]. It uses a region proposal network (RPN) to find regions of interest (RoIs) to refine further. Each RoI in this module has an objectness score, as well as a predicted bounding box. RoIAlign [101] is applied on multi-level feature maps to take RoI information. After passing through a fully connected network, these extracted features are fed to the Transformer encoder as input. The positional info of these RoI proposals is the four values $\left(c_{nx},c_{ny},w_{n},h_{n}\right)$, where $\left(c_{nx},c_{ny}\right)\in {\left[0,1\right]}^{2}$ represents the normalized value of center and $\left(w_{n},h_{n}\right)\in {\left[0,1\right]}^{2}$ represents the normalized value of height and width. Finally, the transformer encoder network inputs these RoIs and their positional encoding for accurate predictions. The FCOS and RCNN modules in TSP-DETR accelerate training convergence and improve the performance of the DETR network.

### 3.7. Conditional-DETR

The cross-attention module in the DETR network needs high-quality input embeddings to predict accurate bounding boxes and class labels. The high-quality content embeddings increase the training convergence difficulty. Conditional-DETR [25] presents a conditional cross-attention mechanism to solve the training convergence issue of DETR. It differs from the simple DETR by input keys $k_{i}$ and input queries $q_{i}$ for cross-attention. In Figure 4, the yellow box represents conditional-DETR. The conditional queries are obtained from 2D coordinates along with the embedding output of the previous decoder layer. The predicted candidate box from decoder-embedding is as follows:(14)$$box=sig(FFN\left(e\right)+{\left[r^{T}00\right]}^{T}).$$

Here, e is the input embedding that is fed as input to the decoder. The box is a 4D vector $\left[box_{cx}box_{cy}box_{w}box_{h}\right]$, having the box center value as $\left(box_{cx},box_{cy}\right)$, width value as $box_{w}$, and height value as $box_{h}$. The sig() function normalizes the predictions and varies from 0 to 1. FFN() predicts the un-normalized box. r is the un-normalized 2D coordinate of the reference-point, and (0,0) is the simple DETR. This work either learns the reference point r for each box or generates it from the respective object query. It learns queries for multi-head cross-attention from input embeddings of the decoder. This spatial query makes the cross-attention head consider the explicit region, which helps to localize the different regions for class labels and bounding boxes by narrowing down the spatial range.

### 3.8. WB-DETR

DETR extracts local features using a CNN backbone and gets global contexts by an encoder–decoder network of the transformer. WB-DETR [26] proves that the CNN backbone for feature extraction in detection transformers is not compulsory. It contains a transformer network without a backbone. It serializes the input image and feeds the local features directly in each independent token to the encoder as input. The transformer self-attention network provides global information, which can accurately obtain the contexts between input image tokens. However, the local features of each token and the information between adjacent tokens need to be included, as the transformer lacks the ability for local feature modeling. The LIE-T2T (Local Information Enhancement-T2T) module solves this issue by reorganizing and unfolding the adjacent patches and focusing on each patch’s channel dimension after unfolding. In Figure 5, the top-right block denotes the LIE-T2T module of WB-DETR. The iterative process of the LIE-T2T module is as follows:(15)P=stretch(reshape(Pi)),(16)$$Q=sig(e_{2}\cdot ReLU\left(e_{1}\cdot P\right)),$$(17)$$P_{i+1}=e_{3}\cdot \left(P\cdot Q\right),$$
where reshape function reorganizes $\left(l_{1}\times c_{1}\right)$ patches into $\left(h_{i}\times w_{i}\times c_{i}\right)$ feature maps. The term stretch denotes unfolding $\left(h_{i}\times w_{i}\times c_{i}\right)$ feature maps to $\left(l_{2}\times c_{2}\right)$ patches. Here, the fully connected layer parameters are $e_{1}$, $e_{2}$, and $e_{3}$. The ReLU activation is its non-linear map function, and the sig generates the final attention. The channel attention in this module provides local information as the relationship between the channels of the patches is the same as the spatial relation in the pixels of the feature maps.

### 3.9. PnP-DETR

The transformer processes the image feature maps that are transformed into a one-dimensional feature vector to produce the final results. Although effective, using the full feature map is expensive because of useless computation on background regions. PnP-DETR [27] proposes a poll and pool (PnP) sampling module to reduce spatial redundancy and make the transformer network computationally more efficient. This module divides the image feature map into contextual background features and fine foreground object features. Then, the transformer network uses these updated feature maps and translates them into the final detection results. In Figure 5, the bottom-left block indicates PnP-DETR. This PnP Sampling module includes two types of samplers: a pool sampler and a poll sampler, as explained below.

**Poll Sampler:** The poll sampler provides fine feature vectors $V_{f}$. A meta-scoring module is used to find the informational value for every spatial location (x, y):(18)$$a_{xy}=ScoreNet\left(v_{xy},\theta s\right).$$

The score value is directly related to the information of feature vector $v_{xy}$. These score values are sorted as follows:(19)$$\left[a_{z},|z=1,\ldots ,Z\right],ℵ=Sort\left(a_{xy}\right),$$
where $Z=h_{i}w_{i}$, and *ℵ* is the sorting order. The top $N_{s}$-scoring vectors are selected to obtain fine features:(20)$$V_{f}=\left[v_{z},|z=1,\ldots ,N_{s}\right].$$

Here, the predicted informative value is considered a modulating factor to sample the fine feature vectors:(21)$$V_{f}=\left[v_{z}\times a_{z},|z=1,\ldots ,N_{s}\right].$$

To make the learning stable, the feature vectors are normalized:(22)$$V_{f}=\left[L_{norm}\left(v_{z}\right)\times a_{z},|z=1,\ldots ,N_{s}\right].$$

Here, $L_{norm}$ is the layer normalization, and $N_{s}=\alpha Z$, where α is the poll ratio factor. This sampling module reduces the training computation.

**Pool Sampler:** The poll sampler obtains the fine features of foreground objects. A pool sampler compresses the background region’s remaining feature vectors that provide contextual information. It performs weighted pooling to get a small number of background features $M_{b}$ motivated by double attention operation [210] and bilinear pooling [211]. The remaining feature vectors of the background region are as follows:(23)$$V_{b}=V\setminus V_{f}=\left\{v_{b},|b=1,\ldots ,Z-N\right\}.$$

The aggregated weights $a_{b}\in R^{M_{b}}$ are obtained by projecting the features with weight values $w^{s}\in R^{c_{i}\times M_{b}}$ as follows:(24)$$a_{b}=v_{b}w^{s}.$$

The projected features with learnable weight $w^{p}\in R^{c_{i}\times c_{i}}$ are obtained as follows:(25)$${\overset{´}{v}}_{b}=v_{b}w^{p}.$$

The aggregated weights are normalized over the non-sampled regions with Softmax as follows:(26)$$a_{bm}=\frac{e_{bm}^{a}}{\sum_{\overset{´}{b}=1}^{N-Z}e^{a}\;\overset{´}{b}\;m}.$$

By using the normalized aggregation weight, the new feature vector is obtained to provide information for non-sampled regions:(27)$$v_{m}=\sum_{b=1}^{Z-N}{\overset{´}{v}}_{b}\times a_{bm}.$$

By considering all Z aggregation weights, the coarse background contextual feature vector is as follows:(28)$$V_{c}=\left\{v_{m},|b=1,\ldots ,M_{b}\right\}.$$

The pool sampler provides contextual information at different scales using aggregation weights. Here, some feature vectors may provide local context while others may capture the global context.

### 3.10. Dynamic-DETR

Dynamic-DETR [28] introduces dynamic attention in the encoder–decoder network of DETR to solve the slow training convergence issue and detection of small objects. Firstly, a convolutional dynamic encoder is proposed to have different attention types to the self-attention module of the encoder network to make the training convergence faster. The attention of this encoder depends on various factors such as spatial effect, scale effect and input feature dimensions effect. Secondly, ROI-based dynamic attention is replaced with cross-attention in the decoder network. This decoder helps to focus on small objects, reduces learning difficulty and converges the network faster. In Figure 5, the bottom right box represents Dynamic-DETR. This dynamic encoder–decoder network is explained in detail as follows.

**Dynamic Encoder:** The Dynamic-DETR uses a convolutional approach for the self-attention module. Given the feature vectors $F=\{F1,\cdot \cdot \cdot ,F_{n}\}$, where n = 5 represents object detectors from the feature pyramid, the multi-scale self-attention (MSA) is as follows:(29)Attn=MSA(F).F.

However, it is impossible because of the various scale feature maps from the FPN. The feature maps of different scales are equalized within neighbors using 2D convolution as in the pyramid convolution [212]. It focuses on the spatial locations of the un-resized mid-layer and transfers information to its scaled neighbors. Moreover, SE [213] is applied to combine the features to provide scale attention.

**Dynamic Decoder:** The dynamic decoder uses mixed attention blocks in place of multi-head layers to ease learning in the cross-attention network and improve the detection of small objects. It also uses dynamic convolution instead of a cross-attention layer inspired by ConvBERT [214] in natural language processing (NLP). Firstly, RoI pooling [1] is introduced in the decoder network, after which position embeddings are replaced with box encoding $BE\in R^{p\times 4}$ as the image size. The output from the dynamic encoder, along with box encoding BE, is fed to the dynamic decoder to pool image features $R\in R^{p\times s\times s\times c_{i}}$ from the feature pyramid as follows:(30)$$R=RoI_{pool}\left(F_{encoder},BE,s\right),$$
where s is the size of the pooling parameter, and $c_{i}$ represents the quantity of channels of $F_{encoder}$. To feed this into the cross-attention module, input embeddings $qe\in R^{p\times c_{i}}$ are required for object queries. These embeddings are passed through the multi-head self-attention (MHSAttn) layer as follows:(31)$$qe^{*}=MHSAttn\left(qe,qe,qe\right).$$

Then, these query embeddings are passed through the fully connected layer (dynamic filters) as follows:(32)$$Filter^{qe}=FC\left(qe^{*}\right).$$

Finally, cross-attention between features and object queries is performed with 1 × 1 convolution using dynamic filters $Filter^{qe}$:(33)$$qe^{F}=Con_{1\times 1}\left(F,Filter^{qe}\right).$$

These features are passed through FFN layers to provide various predictions as updated object-embedding, updated box-encoding, and the object class. This process eases the learning of the cross-attention module by focusing on sparse areas and then spreading to global regions.

### 3.11. YOLOS-DETR

Vision transformer (ViT) [8], inherited from NLP, performs well on the image recognition task. ViT-FRCNN [215] uses a pre-trained backbone (ViT) for a CNN-based detector. It utilizes convolution neural networks and relies on strong 2D inductive biases and region-wise pooling operations for object-level perception. Other similar works, such as DETR [11], introduce 2D inductive bias using CNNs and pyramidal features. YOLOS-DETR [29] presents the transferability and versatility of the transformer from image recognition to detection in the sequence aspect using the least information about the spatial design of the input. It closely follows the ViT architecture with two simple modifications. Firstly, it removes the image-classification patches (CLS) and adds randomly initialized one hundred detection patches (DET) as [216] along with the input patch embeddings for object detection. Secondly, similar to DETR, a bipartite matching loss is used instead of the ViT classification loss. The transformer encoder takes the generated sequence as input as follows:(34)$$\begin{array}{c} s_{0}=\left[I_{p}^{1}L;\cdot \cdot \cdot ;I_{p}^{M}L;I_{d}^{1};\cdot \cdot \cdot ;I_{d}^{100}\right]+\mathrm{PE}, \end{array}$$
where I is the input image $I\in R^{h_{i}\times w_{i}\times c_{i}}$ that is reshaped into 2D tokens $I_{p}\in R^{n_{i}\times \left(R^{2}\cdot c_{i}\right)}$. Here, $h_{i}$ represents the height, and $w_{i}$ indicates the width of the input image. $c_{i}$ is the total number of channels. (r,r) is each token resolution, and $n_{i}=\frac{h_{i}w_{i}}{r^{2}}$ is the total number of tokens. These tokens are mapped to $D_{i}$ dimensions with linear projection, $L\in R^{\left(r^{2}\cdot c_{i}\right)\times D_{i}}$. The result of this projection is $I_{p}L$. The encoder also takes one hundred randomly initialized learnable tokens $I_{d}\in R^{100\times D_{i}}$. To keep the positional information, positional embeddings $\mathrm{PE}\in R^{\left(n_{i}+100\right)\times D_{i}}$ are also added. The encoder of the transformer contains a multi-head self-attention mechanism and one MLP block with a GELU [217] non-linear activation function. Layer normalization (LN) [218] is added between each self-attention and MLP block as follows:(35)$${\overset{´}{s}}_{n}=MHSAttn\left(LN\left(s_{n-1}\right)\right)+s_{n-1},$$(36)$$s_{n}=MLP\left(LN\left({\overset{´}{s}}_{n}\right)\right)+{\overset{´}{s}}_{n},$$
where $s_{n}$ is the encoder input sequence. In Figure 6, the top-right block indicates YOLOS-DETR.

### 3.12. Anchor-DETR

DETR uses learnable embeddings as object queries in the decoder network. These input embeddings do not have a clear physical meaning and cannot illustrate where to focus. It is challenging to optimize the network as object queries concentrate on something other than specific regions. Anchor-DETR [30] solves this issue by proposing object queries as anchor points that are extensively used in CNN-based object detectors. This query design can provide multiple object predictions in one region. Moreover, a few modifications in the attention are proposed that reduce the memory cost and improve performance. In Figure 6, the yellow block shows Anchor-DETR. The two main contributions of Anchor-DETR, query and attention variant design, are explained as follows.

**Row and Column Decoupled-Attention:** DETR requires huge GPU memory, as in [219,220], because of the complexity of the cross-attention module. It is more complex than the self-attention module in the decoder. Although Deformable-DETR reduces memory cost, it still causes random memory access, making the network slower. Row–column decoupled attention (RCDA), as shown in the blue block of Figure 6, reduces memory and provides similar or better efficiency.

**Anchor Points as Object Queries:** The CNN-based object detectors consider anchor points as the relative position of the input feature maps. In contrast, transformer-based detectors take uniform grid locations, handcraft locations, or learned locations as anchor points. Anchor-DETR considers two types of anchor points: learned anchor locations and grid anchor locations. The grid anchor locations are input image grid points. The learned anchor locations are uniform distributions from 0 to 1 (randomly initialized) and updated using the learned parameters.

### 3.13. Sparse-DETR

Sparse-DETR [31] filters the encoder tokens by a learnable cross-attention map predictor. After distinguishing these tokens in the decoder network, it focuses only on foreground tokens to reduce computational costs.

Sparse-DETR introduces the scoring module, aux-heads in the encoder, and the Top-k queries selection module for the decoder. In Figure 6, the light orange box represents Sparse-DETR. Firstly, it determines the saliency of tokens, fed as input to the encoder, using the scoring network that selects top ρ% tokens. Secondly, the aux-head takes the top-k tokens from the output of the encoder network. Finally, the top-k tokens are used as the decoder object queries. The salient token prediction module refines encoder tokens that are taken from the backbone feature map using threshold ρ and updates the features $x_{l}-1$ as follows:$$x_{l}^{m}=\left\{\begin{array}{cc} x_{l-1}^{m} & \;m\notin \Omega_{r}^{q}\\ \mathrm{LN}(\mathrm{FFN}\left(y_{l}^{m}\right)+y_{l}^{m}) & \;m\in \Omega_{r}^{q}, \end{array}\right.,$$(37)$$where\;y_{l}^{m}=LN\left(DeformAttn\left(x_{l-1}^{m},x_{l-1}\right)+x_{l-1}^{m}\right),$$
where DeformAttn is the deformable attention, FFN is the feed-forward network, and LN is the layer normalization. Then, the decoder cross-attention map (DAM) accumulates the attention weights of decoder object queries, and the network is trained by minimizing loss between prediction and binarized DAM as follows:(38)$$L_{dam}=\frac{-1}{M}\sum_{k=1}^{M}BCELoss\left(sn\left(x_{f}\right),DAM_{k}^{b}\right),$$
where BCELoss is the binary cross-entropy (BCE) loss, $DAM_{k}^{b}$ is the k-th binarized DAM value of the encoder token, and sn is the scoring network. In this way, sparse-DETR minimizes the computation by significantly eliminating encoder tokens.

### 3.14. D^2^ETR

Much work [20,22,23,24,25] has been proposed to make the training convergence faster by modifying the cross-attention module. Many researchers [20] used multi-scale feature maps to improve performance for small objects. However, the solution for high computation complexity has yet to be proposed. D^2^ETR [32] achieves better performance with low computational cost. Without an encoder module, the decoder directly uses the fine-fused feature maps provided by the backbone network with a novel cross-scale attention module. The D^2^ETR contains two main modules: a backbone and a decoder. The backbone network based on a pyramid vision transformer (PVT) consists of two parallel layers: one for cross-scale interaction and another for intra-scale interaction. This backbone contains four transformer levels to provide multi-scale feature maps. All levels have the same architecture, depending on the basic module of the selected transformer. The backbone also contains three fusing levels in parallel with four transformer levels. These fusing levels provide a cross-scale fusion of input features. The i-th fusing level is shown in the light green block of Figure 7. The cross-scale attention is formulated as follows:(39)$$f_{j}=L_{j}\left(f_{j-1}\right),$$(40)$$f_{j}^{*}=SA\left(f_{q},f_{k},f_{v}\right),$$(41)$$f_{q}=f_{j},f_{k}=f_{v}=\left[f_{1}^{*},f_{2}^{*},\ldots ,f_{j-1}^{*},f_{j}\right],$$
where $f_{j}^{*}$ is the fused form feature map $f_{j}$. Given that L is the input of the decoder as the last-level feature map, the final result of cross-scale attention is $f_{1}^{*},f_{2}^{*},\ldots ,f_{L}^{*}$. The output of this backbone is fed to the decoder that takes object queries in parallel. It provides output embeddings independently transformed into class labels and box coordinates by a forward feed network. Without an encoder module, the decoder directly used the fine-fused feature maps provided by the backbone network, with a novel cross-scale attention module providing better performance with low computational cost.

### 3.15. FP-DETR

Modern CNN-based detectors like YOLO [221] and Faster-RCNN [1] utilize specialized layers on top of backbones pre-trained on ImageNet to enjoy pre-training benefits such as improved performance and faster training convergence. The DETR network and its improved version [21] only pre-train its backbone while training both encoder and decoder layers from scratch. Thus, the transformer needs massive training data for fine-tuning. The main reason for not pre-training the detection transformer is the difference between the pre-training and final detection tasks. Firstly, the decoder module of the transformer takes multiple object queries as input for detecting objects, while ImageNet classification takes only a single query (class token). Secondly, the self-attention module and the projections on input query embeddings in the cross-attention module easily overfit a single class query, making the decoder network difficult to pre-train. Moreover, the downstream detection task focuses on classification and localization, while the upstream task considers only classification for the objects of interest.

FP-DETR [33] reformulates the pre-training and fine-tuning stages for detection transformers. In Figure 7, the pink block indicates FP-DETR. It takes only the encoder network of the detection transformer for pre-training, as it is challenging to pre-train the decoder on the ImageNet classification task. Moreover, DETR uses both the encoder and CNN backbone as feature extractors. FP-DETR replaces the CNN backbone with a multi-scale tokenizer and uses the encoder network to extract features. It fully pre-trains the Deformable-DETR on the ImageNet dataset and fine-tunes it for final detection that achieves competitive performance.

### 3.16. CF-DETR

CF-DETR [34] observes that COCO-style metric average precision (AP) results for small objects on detection transformers at low IoU threshold values are better than CNN-based detectors. It refines the predicted locations by utilizing local information, as incorrect bounding box location reduces performance on small objects. CF-DETR introduces the transformer-enhanced FPN (TEF) module, coarse layers, and fine layers into the decoder network of DETR. In Figure 7, the blue box represents CF-DETR. The TEF module provides the same functionality as FPN, has non-local features E4 and E4 extracted from the backbone, and E5 features taken from the encoder output. The features of the TEF module and the encoder network are fed to the decoder as input. The decoder modules introduce a coarse block and a fine block. The coarse block selects foreground features from the global context. The fine block has two modules: adaptive scale-fusion (ASF) and local cross-attention (LCA), further refining coarse boxes. In summary, these modules refine and enrich the features by fusing global and local information to improve detection transformer performance.

### 3.17. DAB-DETR

DAB-DETR [72] uses the bounding box coordinates as object queries in the decoder and gradually updates them in every layer. In Figure 8, the purple block indicates DAB-DETR. These box coordinates make training convergence faster by providing positional information and using the height and width values to update the positional attention map. This type of object query provides better spatial information prior to the attention mechanism and provides a simple query formulation mechanism.

The decoder network contains two main networks: a self-attention network to update queries and a cross-attention network to find feature probing. The difference between the self-attention of the original DETR and DAB-DETR is that the query and key matrices also have position information taken from bounding box coordinates. The cross-attention module concatenates the position and content information in key and query matrices and determines their corresponding heads. The decoder takes input embeddings as content queries and anchor boxes as positional queries to find object probabilities related to anchors and content queries. This way, dynamic box coordinates used as object queries provide better prediction, making the training convergence faster and increasing detection results for small objects.

### 3.18. DN-DETR

DN-DETR [35] uses noised object queries as an additional decoder input to reduce the instability of the bipartite-matching mechanism in DETR, which causes the slow convergence problem. In Figure 8, the dark green block indicates DN-DETR. The decoder queries have two parts: the denoising part, containing noised ground-truth box-label pairs as input, and the matching part, containing learnable anchors as input. The matching part $M=\{M_{0},M_{1},\ldots ,M_{l-1}\}$ determines the resemblance between the ground-truth label pairs and the decoder output, while the denoising part $d=\{d_{0},d_{1},\ldots ,d_{k-1}\}$ attempts to reconstruct the ground-truth objects as follows:(42)Output=Decoder(d,M,I|A),
where *I* is the image features taken as input from the transformer encoder, and *A* is the attention mask that stops the information transfer between the matching and denoising parts and among different noised levels of the same ground-truth objects. The decoder has noised levels of ground-truth objects where noise is added to bounding boxes and class labels, such as label flipping. It contains a hyperparameter λ for controlling the noise level. The training architecture of DN-DETR is based on DAB-DETR, as it also takes bounding box coordinates as object queries. The only difference between these two architectures is the class label indicator as an additional input in the decoder to assist label denoising. The bounding boxes are updated inconsistently in DAB-DETR, making relative offset learning challenging. The denoising training mechanism in DN-DETR improves performance and training convergence.

### 3.19. AdaMixer

AdaMixer [36] considers the encoder an extra network between the backbone and decoder that limits the performance and slows the training convergence because of its design complexity. AdaMixer provides a detection transformer network without an encoder. In Figure 8, the light green box represents AdaMixer. The main modules of AdaMixer are explained as follows.

**Three-dimensional feature space:** For the 3D feature space, the input feature map from the CNN backbone with the downsampling stride $s_{i}^{f}$ is first transformed by a linear layer to the same $d_{f}$ channel and computed the coordinate of its z-axis as follows:(43)$$z_{i}^{f}=log_{2}\left(s_{i}^{f}/s_{b}\right),$$
where the height $h_{i}$ and width $w_{i}$ of feature maps (different strides) is rescaled to $h_{i}/s_{b}$ and $w_{i}/s_{b}$, where $s_{b}=4$.

**Three-dimensional feature-sampling process:** In the sampling process, the query generates $I_{p}$ groups of vectors to $I_{p}$ points, $\left(\Delta x_{j},\Delta y_{j},\Delta z_{j}\right)I_{p}$, where each vector is dependent on its content vector $q_{i}$ by a linear layer $L_{i}$ as follows:(44)$$\left(\Delta x_{j},\Delta y_{j},\Delta z_{j}\right)I_{p}=L_{i}\left(q_{i}\right).$$

These offset values are converted into sampling positions with regard to the position vector of the object query as follows:(45)$$\left\{\begin{array}{cc} \tilde{x_{j}} & =x+\Delta x_{j}.2^{z-r},\\ \tilde{y_{j}} & =y+\Delta y_{j}.2^{z+r},\\ \tilde{z_{j}} & =z+\Delta z_{j}. \end{array}\right.$$

The interpolation over the 3D feature space first samples by bilinear interpolation in the $\left(x_{i},y_{i}\right)$ space and then interpolates on the z-axis by Gaussian weighting, where the weight for the i-th feature map is as follows:(46)$${\tilde{w}}_{i}=\frac{exp(-{\left(\tilde{z}-z_{i}^{f}\right)}^{2}/\Gamma_{z})}{\sum_{i}exp\left(-{\left(\tilde{z}-z_{i}^{f}\right)}^{2}/\Gamma_{z}\right)},$$
where $\Gamma_{z}$ is the softening coefficient used to interpolate values over the z-axis ($\Gamma_{z}=2$ ). This process makes decoder detection learning easier by taking feature samples according to the query.

**AdaMixer Decoder:** The decoder module in AdaMixer takes a content vector $q_{i}$ and positional vector $\left(x_{i},y_{i},z_{i},r_{i}\right)$ as input object queries. The position-aware multi-head self-attention is applied between these queries as follows:(47)$$Attn\left(q_{i},k_{i},v_{i}\right)=Softmax\left(\frac{q_{i}k_{i}^{T}}{\sqrt{d}}+\alpha X\right).v_{i},$$
where $X_{kl}=log(\left|box_{k}\cap box_{l}\left|box_{k}\right|+\epsilon \right),\epsilon =10^{-7}$. The $X_{kl}=0$ indicates the $box_{k}$ is inside the $box_{l}$ and $X_{kl}=l$ represents no overlapping between $box_{k}$ and $box_{l}$. This position vector is updated at every stage of the decoder network. The AdaMixer decoder module takes a content vector and a positional vector as input object queries. For this, multi-scale features taken from the CNN backbone are converted into a 3D feature space, as the decoder should consider $\left(x_{i},y_{i}\right)$ space as well as be adjustable in terms of scales of detected objects. It takes the sampled features from this feature space as input. It applies the AdaMixer mechanism to provide final predictions of input queries without using an encoder network to reduce the computational complexity of detection transformers.

### 3.20. REGO-DETR

REGO-DETR [37] proposes an RoI-based method for detection refinement to improve the attention mechanism in DETR. In Figure 9, the purple color block denotes REGO-DETR. It contains two main modules: a multi-level recurrent mechanism and a glimpse-based decoder. In the multi-level recurrent mechanism, bounding boxes detected in the previous level are considered to get glimpse features. These are converted into refined attention using earlier attention in describing objects. The k-th processing level is as follows:(48)$$\left\{\begin{array}{c} O_{class}\left(k\right)=DF_{class}\left(H_{de}\left(k\right)\right),\\ O_{bbox}\left(k\right)=DF_{bbox}\left(H_{de}\left(k\right)\right)+O_{bbox}\left(k-1\right), \end{array}\right.$$
where $O_{class}\in R^{M_{d}\times M_{c}}$ and $O_{bbox}\in R^{M_{d}\times 4}$. Here, $M_{d}$ and $M_{c}$ represent the total number of predicted objects and classes, respectively. $DF_{class}$ and $DF_{bbox}$ are functions that convert the input features into desired outputs. $H_{de}\left(k\right)$ is the attention of this level after decoding as follows:(49)$$H_{de}\left(k\right)=\left[H_{gm}\left(k\right),H_{de}\left(k-1\right)\right],$$
where $H_{gm}\left(k\right)$ refers to the glimpse features according to $H_{de}\left(k-1\right)$ and previous levels. These glimpse features are transformed using multi-head cross-attention into refined attention outputs according to previous attention outputs as follows:(50)$$H_{gm}\left(k\right)=Attn\left(V\left(k\right),H_{de}\left(k-1\right)\right).$$
For extracting glimpse features V(k), the following operation is performed:(51)$$V\left(k\right)=FE_{ext}\left(X,RI\left(O_{bbox}\left(k-1\right),\alpha \left(k\right)\right)\right),$$
where $FE_{ext}$ is the feature extraction function, α(k) is a scalar parameter, and RI is the RoI computation. In this way, region of interest (RoI)-based refinement modules make the training convergence of the detection transformer faster and provide better performance.

### 3.21. DINO

DN-DETR adds positive noise to the anchors taken as object queries to the input of the decoder and provides labels to only those anchors with ground-truth objects nearby. Following DAB-DETR and DN-DETR, DINO [38] proposes a mixed object query selection method for anchor initialization and a look forward twice mechanism for box prediction. It provides the contrastive denoising (CDN) module, which takes positional queries as anchor boxes and adds additional DN loss. In Figure 9, the red block indicates DINO. This detector uses $\lambda_{1}$ and $\lambda_{2}$ hyperparameters where $\lambda_{1}<\lambda_{2}$. The bounding box $b=(x_{i},y_{i},w_{i},h_{i})$ taken as input in the decoder, its corresponding generated anchor is denoted as $a=(x_{i},y_{i},w_{i},h_{i})$.(52)$$ATD\left(k\right)=\frac{1}{k}\Sigma \{M_{K}\left(\{\right\|b_{0}-a_{0}{\|}_{1},\|b_{1}-a_{1}{\|}_{1},\ldots ,\|b_{N-1}-a_{N-1}{\|}_{1}\},k)\},$$
where $\left\|(b_{i}-a_{i})\right\|$ is the distance between the anchor and bounding box, and $M_{K}\left(x,k\right)$ is the function that provides the top K elements in x. The λ parameter is the threshold value for generating noise for anchors that are fed as input object queries to the decoder. It provides two types of anchor queries: positive with threshold value less than $\lambda_{1}$ and negative with noise threshold values greater than $\lambda_{1}$ and less than $\lambda_{2}$. This way, the anchors with no ground-truth nearby are labeled as “no object”. Thus, DINO makes the training convergence faster and improves performance for small objects.

DINOv2 [222] is a self-supervised vision transformer model developed by Meta AI. It was trained on a large-scale dataset of 142 million images without any labels or annotations. DINOv2 [222] produces high-performance visual features that can be directly employed with classifiers as simple as linear layers on a variety of computer vision tasks. These visual features are robust and perform well across domains without any requirement for fine-tuning. DINOv3 [223], also developed by Meta AI, is the third generation of the DINO framework. It is a 7-billion-parameter Vision Transformer trained on 1.7 billion images without labels. DINOv3 [223] introduces several innovations, including Gram anchoring, which stabilizes dense feature maps during training, and axial RoPE (Rotary Positional Embeddings) with jittering, which enhances the model’s robustness to varying image resolutions, scales, and aspect ratios. These advancements enable DINOv3 [223] to achieve state-of-the-art performance across a wide range of vision tasks, including object detection, semantic segmentation, and depth estimation.

### 3.22. Co-DETR

Co-DETR [39] is an improvement over DETR that addresses a key limitation of the standard one-to-one label assignment, which in DETR restricts each ground-truth object to a single predicted query. In Figure 10, the light red block indicates Co-DETR. This design leads to a few positive samples during training, leaving many decoder queries unused and slowing gradient flow, particularly in the early stages of learning. Co-DETR overcomes this by introducing a collaborative hybrid assignment strategy that combines the original one-to-one assignment with a one-to-many assignment implemented through auxiliary heads. The one-to-one assignment preserves the unique matching of each object, maintaining the stability and structure of DETR’s training. The one-to-many assignment leverages heuristics from classical object detectors, such as ATSS or Faster R-CNN, to assign multiple predicted queries to the same ground-truth object, providing denser supervision for both the encoder and decoder. The auxiliary heads are only active during training and are discarded during inference, ensuring no additional computational cost at test time.

The total training loss is expressed as follows:(53)$$L_{\mathrm{total}}=L_{\mathrm{DET}}+\sum_{h\in \mathrm{auxiliary}\;\mathrm{heads}}L_{\mathrm{aux},h},$$
where $L_{\mathrm{DET}}$ is the standard DETR loss and $L_{\mathrm{aux},h}$ represents the one-to-many assignment loss from each auxiliary head. This hybrid assignment improves gradient flow by increasing the number of positive samples per batch, enhances encoder supervision through additional feedback signals, and leads to better detection performance on benchmarks such as COCO and LVIS. By enriching training supervision without altering the inference process, Co-DETR enables faster convergence, more effective learning, and higher accuracy in DETR-based object detectors.

### 3.23. LW-DETR

LW-DETR [40] is a lightweight, transformer-based object detection model designed for high accuracy and real-time performance. It streamlines the standard DETR architecture by using an optimized vision transformer (ViT) encoder and a shallow decoder. The model first processes an input image by breaking it into patches and extracting features through the encoder. These features are then refined via a convolutional projection layer before being passed to the decoder, which uses a set of object queries to predict bounding boxes and class labels. In Figure 10, the blue block indicates LW-DETR. LW-DETR further improves efficiency through several strategies: interleaved window and global attention reduce the complexity of self-attention, multi-level feature aggregation captures richer representations, and window-major feature map organization optimizes attention computation. During training, the model employs deformable cross-attention to focus on relevant regions, IoU-aware classification loss to enhance localization accuracy, and encoder–decoder pre-training to learn robust features. The total training loss combines classification, bounding box regression, and IoU losses to guide learning effectively.(54)$$L_{\mathrm{total}}=L_{\mathrm{cls}}+L_{\mathrm{box}}+\lambda_{\mathrm{giou}}L_{\mathrm{GIoU}},$$
where $L_{\mathrm{cls}}$ is the classification loss, $L_{\mathrm{box}}$ is the bounding box regression loss, $L_{\mathrm{GIoU}}$ is the generalized intersection over union loss, and $\lambda_{\mathrm{giou}}$ balances the contributions of the losses. Experimental results show that LW-DETR achieves higher accuracy than many real-time detectors, including YOLO variants, while maintaining low computational cost, making it suitable for real-time object detection tasks.

### 3.24. RT-DETR

An RT-DETR [41] (real-time detection transformer) is a transformer-based object detection model developed by Baidu, designed for high-speed, end-to-end inference suitable for real-time applications. In Figure 10, the purple block indicates RT-DETR. The model employs a hybrid encoder that processes multi-scale features by decoupling intra-scale interactions and cross-scale feature fusion. This efficient design reduces computational costs while retaining rich feature representations. The encoder outputs multi-scale feature maps, which are then passed to a DETR-style decoder. An IoU-aware query selection mechanism is utilized to focus on the most relevant object queries, enhancing detection accuracy. Additionally, the inference speed can be adjusted by changing the number of decoder layers, allowing for flexible deployment across different real-time scenarios.

Subsequent versions build upon this foundation to further enhance performance. RT-DETRv2 [224] introduces selective multi-scale sampling and replaces the grid-sample operator with a discrete sampling operator, improving the detection of objects at different scales. It also employs dynamic data augmentation and scale-adaptive hyperparameter tuning to enhance training efficiency without increasing inference latency. RT-DETRv3 [225] addresses limitations of sparse supervision and insufficient decoder training by adding a CNN-based auxiliary branch for dense supervision, a self-attention perturbation strategy to diversify label assignment, and a shared-weight decoder branch for dense positive supervision. In summary, the RT-DETR series demonstrates a clear evolution in real-time object detection, with each version introducing architectural and training innovations that enhance both speed and accuracy. The original RT-DETR establishes the foundation for real-time performance, while v2 and v3 progressively improve detection capability without compromising inference efficiency.

It is important to compare modifications in detection transformers to understand their effect on network size, training convergence, and performance. In this work, we use the COCO2014 mini validation set (minival) as a benchmark, since COCO is a widely accepted standard for evaluating object detection models [75]. All images are preprocessed using standard resizing and normalization procedures, and data augmentation, such as random horizontal flipping, is applied, consistent with typical DETR training protocols. The performance of DETR and its variants is evaluated using mean average precision (mAP), calculated as the mean of each object category’s average precision (AP), where AP corresponds to the area under the precision–recall curve [226]. Following the standard COCO evaluation protocol, objects are classified into three size categories based on pixel area: small (<$32^{2}$ pixels), medium ($32^{2}$–$96^{2}$ pixels), and large (>$96^{2}$ pixels). This categorization allows for detailed analysis across object scales, with AP_S_, AP_M_, and AP_L_ reporting performance for small, medium, and large objects, respectively. For a fair comparison, all results are obtained by loading the original pre-trained PTH files released by the respective authors and validating them on the COCO minival set. This approach allows us to reproduce the reported performance of each model while focusing on the architectural differences and improvements introduced by various DETR variants.

## 4. Results and Discussion

Many advancements are proposed in DETR, such as backbone modification, query design, and attention refinement to improve performance and training convergence. Table 3 shows the performance comparison of all DETR-based detection transformers on the COCO minival set. We can observe that DETR performs well at 500 training epochs and has low AP on small objects. The modified versions improve performance and training convergence, like DINO, which has an mAP of 49.0% at 12 epochs and performs well on small objects.

The quantitative analysis of DETR and its updated versions regarding training convergence and model size on the COCO minival set is performed. Left side of Figure 11 shows the mAP of the detection transformers using a ResNet-50 backbone with training epochs. The original DETR, represented with a brown line, has low training convergence. It has an mAP value of 35.3% at 50 training epochs and 44.9 % at 500 training epochs. Here, DINO, represented with a red line, converges at low training epochs and gives the highest mAP on all epoch values. The attention mechanism in DETR involves computing pairwise attention scores between every pair of feature vectors, which can be computationally expensive, especially for large input images. Moreover, the self-attention mechanism in DETR relies on using fixed positional encodings to encode the spatial relationships between the different parts of the input image. This can slow down the training process and increase convergence time. In contrast, Deformable-DETR and DINO have some modifications that can help speed up the training process. For example, Deformable DETR introduces deformable attention layers, which can better capture spatial context information and improve object detection accuracy. Similarly, DINO uses a denoising learning approach to train the network to learn more generalized features useful for object detection, making the training process faster and more effective.

Right side of Figure 11 compares all detection transformers regarding the model size. Here, YOLOS-DETR uses DeiT-small as the backbone instead of DeiT-Ti, but it also increases the model size by 20x times. DINO and REGO-DETR have comparable mAP, but REGO-DETR is nearly double the model size of DINO. These networks use more complex architectures than the original DETR architecture, which increases the total parameters and the overall network size.

We also provide a qualitative analysis of DETR and its updated versions on all-sized objects in Figure 12. For small objects, the mAP for the original DETR is 15.2% at 50 epochs, while Deformable-DETR has an mAP value of 26.4% at 50 epochs. The self-attention mechanism in Deformable-DETR allows it to interpolate features from neighboring pixels, which is particularly useful for small objects that may only occupy a few pixels in an image. This mechanism in Deformable-DETR captures more precise and detailed information about small objects, which can lead to better performance than DETR.

While DINO demonstrates impressive accuracy and fast convergence, its computational footprint remains a significant concern. With approximately 860 GFLOPs per inference, DINO is far more demanding than lightweight alternatives such as Nano YOLO variants, which typically operate in the range of 5–10 GFLOPs. This stark difference highlights a fundamental limitation of many DETR-based models: despite their accuracy gains, their inference cost makes them impractical for latency-critical or resource-constrained applications. In contrast, RT-DETR and LW-DETR provide lightweight and real-time DETR variants, achieving competitive accuracy with a substantially lower computational load (136–259 GFLOPs for RT-DETR and 67.7 GFLOPs for LW-DETR). Additionally, Co-DETR focuses on enhancing contextual reasoning to further boost detection performance, achieving very high AP scores, though at a higher computational cost similar to DINO. Thus, future research must address not only accuracy and convergence speed but also the efficiency gap that separates DETR variants from lightweight CNN-based detectors, ensuring their practical applicability in real-world scenarios.

While Table 3 and Figure 11 and Figure 12 show performance improvements, it is also important to consider computational cost, memory footprint, and implementation complexity. Models like DINO and REGO achieve high mAP but require significantly more parameters and GFLOPs, making them less suitable for resource-constrained scenarios. Deformable-DETR provides a balanced trade-off by improving small object detection and convergence speed without drastically increasing computational load. YOLOS-DETR, while compact in design, relies on a transformer backbone (DeiT-S) that increases the memory requirement by up to 20×, highlighting a trade-off between model size and detection speed. Therefore, selecting a DETR variant depends not only on accuracy but also on hardware constraints, dataset characteristics, and real-time requirements.

## 5. Open Challenges and Future Directions

Detection transformers have shown promising results on various object detection benchmarks. However, several open challenges remain, providing directions for future improvements. Table 4 summarizes the advantages and limitations of the various improved versions of DETR. Some of the key open challenges and future directions are as follows.

**Improving the attention mechanisms:** The performance of detection transformers heavily relies on the attention mechanism to capture dependencies between spatial locations in an image. To date, around 60% of modifications in DETR have focused on the attention mechanism to improve performance and training convergence. Future research could explore more refined attention mechanisms to better capture spatial information or incorporate task-specific constraints.

**Adaptive and dynamic backbones:** The backbone architecture significantly affects network performance and size. Current detection transformers often use fixed backbones or remove them entirely. Only about 10% of DETR modifications have targeted the backbone to improve performance or reduce model size. Future work could investigate dynamic backbone architectures that adjust their complexity based on the input image, potentially enhancing both efficiency and accuracy.

**Improving the quantity and quality of object queries:** In DETR, the number of object queries fed to the decoder is typically fixed during training and inference, but the number of objects in an image varies. Later approaches, such as DAB-DETR, DN-DETR, and DINO, demonstrate that adjusting the quantity or quality of object queries can significantly impact performance. DAB-DETR uses dynamic anchor boxes as queries, DN-DETR adds positive noise to queries for denoising training, and DINO adds both positive and negative noise for improved denoising. Future models could dynamically adjust the number of object queries based on image content and incorporate adaptive mechanisms to improve query quality.

**Emerging directions:** Beyond attention mechanisms, backbones, and object queries, several additional challenges remain. Improving training efficiency through faster convergence strategies and sample-efficient learning could make DETR more practical for large-scale applications. Integrating multitask learning, such as jointly performing detection, segmentation, and tracking, can leverage shared representations for better performance. Enhancing robustness and generalization under occlusions, domain shifts, or low-resolution inputs is also critical. Interdisciplinary approaches could incorporate reinforcement learning to guide model adaptation, NLP-inspired sequence modeling to improve feature interactions, or graph-based reasoning techniques to capture relationships between objects. Concrete research challenges include designing models that dynamically adapt to new tasks or domains and developing cross-modal attention mechanisms that integrate multiple data sources for richer scene understanding.

## 6. Conclusions

Detection transformers have transformed object detection by enabling fully end-to-end models that eliminate the need for proposal generation and complex post-processing, while also providing insights into the inner workings of deep neural networks. This review presented a detailed overview of DETR and its variants, focusing on recent advancements designed to improve performance and training convergence. In particular, modifications to the attention module in the encoder–decoder network and updates to object queries have enhanced training stability and performance, especially for small objects. Other improvements include backbone refinements, query design enhancements, and attention mechanism optimizations, all of which contribute to better accuracy and efficiency. From this survey, several high-level patterns emerge. Slow convergence and limited small-object detection remain central challenges, driving innovations in attention mechanisms, query design, and backbone architecture. Across DETR variants, commonalities include the use of transformer-based attention, modular encoder–decoder design, and strategies to increase positive supervision, while differences arise in how variants balance accuracy versus efficiency, implement multi-scale feature fusion, and assign object queries. Research diverges along two primary paths: accuracy-focused methods leverage deeper backbones and extensive pre-training, while efficiency-oriented approaches adopt lightweight, sparse, or deformable architectures such as RT-DETR and LW-DETR, which achieve competitive performance with lower computational cost. Recent trends further emphasize efficiency, multitask learning, and cross-modal integration, enabling faster convergence, improved generalization, and broader scene understanding that encompasses detection, segmentation, tracking, and vision–language reasoning. Key insights from this survey indicate that model design is increasingly shaped by the trade-off between real-time deployment and high accuracy, and that modular, adaptive architectures are central to achieving this balance. Overall, DETR has evolved into a modular and flexible framework capable of balancing accuracy and efficiency. Future directions point toward adaptive architectures that dynamically allocate computational resources based on input complexity, robust training strategies for challenging environments, and richer contextual reasoning through multimodal integration. By uniting architectural innovation with practical deployment considerations, transformers are poised to drive the next generation of scalable, intelligent, and versatile visual perception systems.

## Figures and Tables

**Figure 1 sensors-25-06025-f001:**
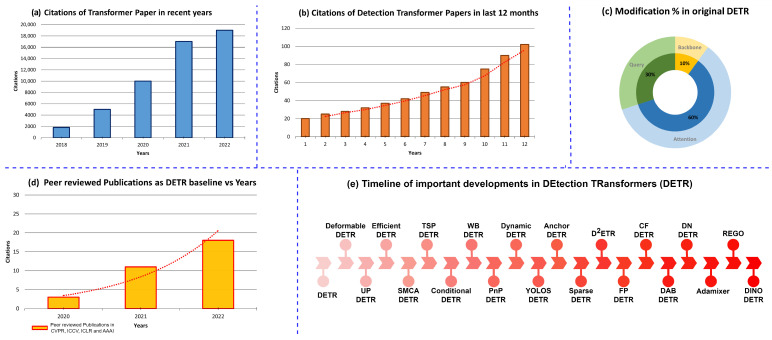
Statistical overview of the literature on transformers. (**a**) Number of citations per year for transformer papers. (**b**) Citations in the last 12 months on detection transformer papers. (**c**) Modification percentage in the original detection transformer (DETR) to improve the performance and convergence speed. (**d**) Number of peer-reviewed publications per year that used DETR as a baseline. (**e**) A non-exhaustive timeline overview of important developments in DETR for detection tasks.

**Figure 2 sensors-25-06025-f002:**
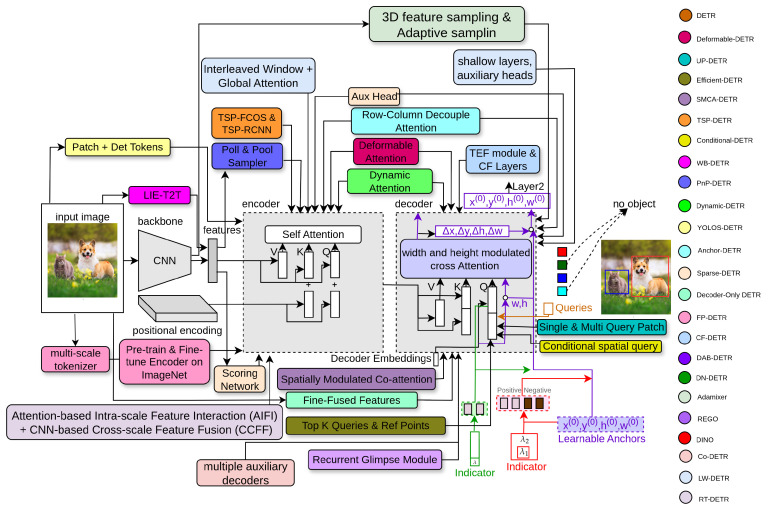
An overview of the detection transformer (DETR) and its modifications proposed by recent methods to improve performance and training convergence. It considers the detection a set prediction task and uses the transformer to free the network from post-processing steps such as non-maximal suppression (NMS). Here, each module added to the DETR is represented by different color with its corresponding label (shown on the right side).

**Figure 3 sensors-25-06025-f003:**
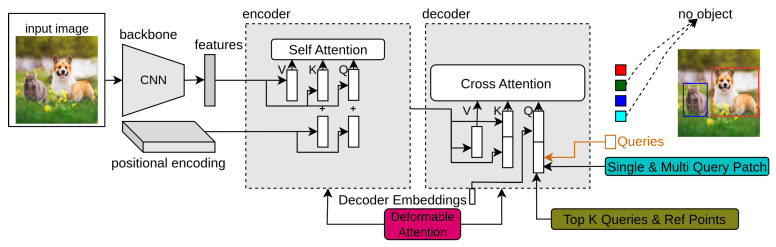
The structure of the original DETR after the addition of Deformable-DETR [20], UP-DETR [21], and Efficient-DETR [22]. Here, the network is a simple DETR network, along with improvement indicated by small colored boxes. The dark pink block indicates Deformable-DETR, the bright cyan block indicates UP-DETR, and the dull green box represents Efficient-DETR.

**Figure 4 sensors-25-06025-f004:**
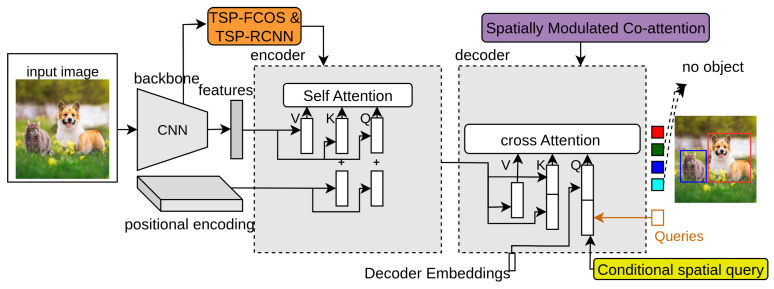
The structure of the original DETR after the addition of SMCA-DETR [23], TSP-DETR [24], and Conditional-DETR [25]. Here, the network is a simple DETR network, along with improvement indicated with small colored boxes. The purple block indicates SMCA-DETR, the orange block indicates TSP-DETR, and the yellow box represents Conditional-DETR.

**Figure 5 sensors-25-06025-f005:**
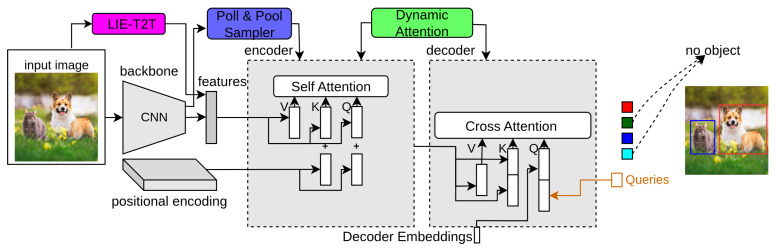
The structure of the original DETR after the addition of WB-DETR [26], PnP-DETR [27], and Dynamic-DETR [28]. Here, the network is a simple DETR network, along with improvement indicated with small colored boxes. The Magenta block indicates WB-DETR, the blue block indicates PnP-DETR, and the green box represents Dynamic-DETR.

**Figure 6 sensors-25-06025-f006:**
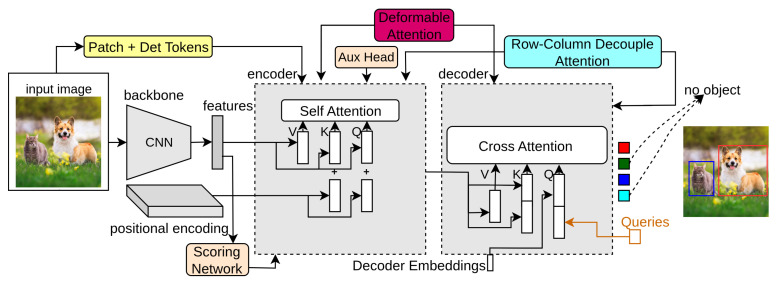
The structure of the original DETR after the addition of YOLOS-DETR [29], Anchor-DETR [30], and Sparse-DETR [31]. Here, the network is a simple DETR network, along with improvement indicated with small colored boxes. The yellow block indicates YOLOS-DETR, the light blue block indicates Anchor-DETR, and the light orange box represents Sparse-DETR.

**Figure 7 sensors-25-06025-f007:**
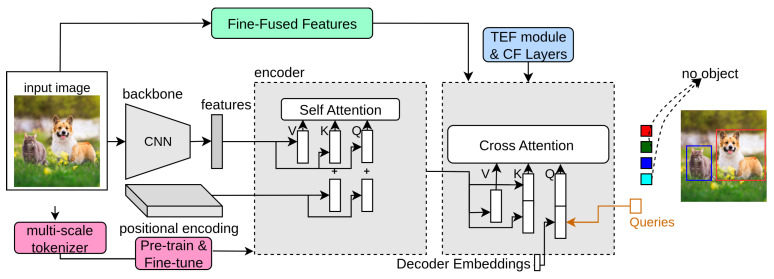
The structure of the original DETR after the addition of D^2^ETR [32], FP-DETR [33], and CF-DETR [34]. Here, the network is a simple DETR network, along with improvement indicated with small colored boxes. The light green block indicates D^2^ETR, the pink block indicates FP-DETR, and the blue box represents CF-DETR.

**Figure 8 sensors-25-06025-f008:**
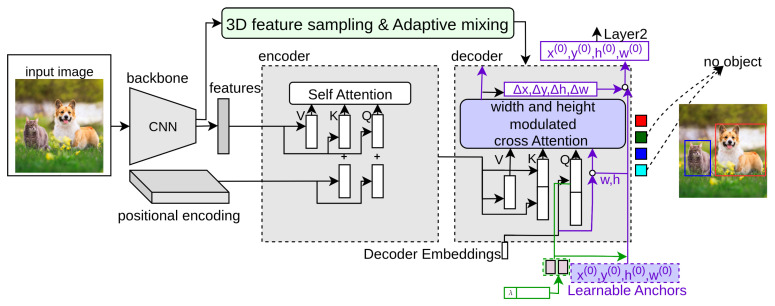
The structure of the original DETR after the addition of DAB-DETR [72], DN-DETR [35], and AdaMixer [36]. Here, the network is a simple DETR network, along with improvement indicated with small colored boxes. The purple block indicates DAB-DETR, the dark green block indicates DN-DETR, and light green box represents AdaMixer.

**Figure 9 sensors-25-06025-f009:**
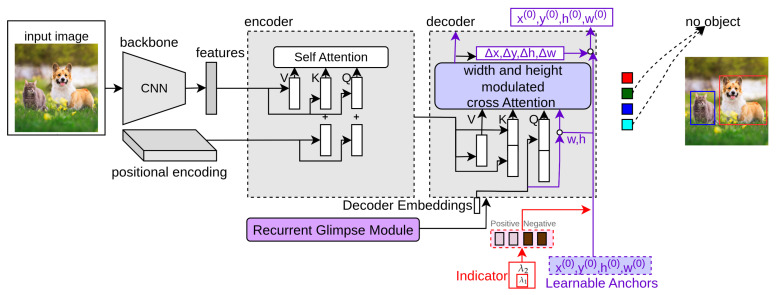
The structure of the original DETR after the addition of REGO-DETR [37] and DINO [38]. Here, the network is a simple DETR network, along with improvement indicated with small colored boxes. The purple color block indicates REGO-DETR and the red indicator block represents DINO.

**Figure 10 sensors-25-06025-f010:**
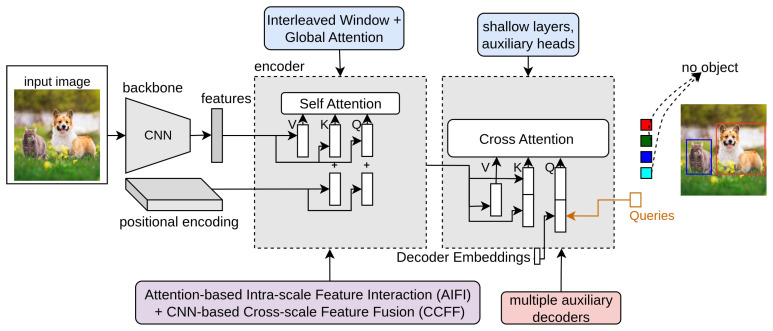
The structure of the original DETR after the addition of Co-DETR [37], RT-DETR, and LW-DETR [38]. Here, the network is a simple DETR network, along with improvement indicated with small colored boxes. The light red indicator block represents Co-DETR, blue color block indicates LW-DETR, and the purple color block indicates RT-DETR.

**Figure 11 sensors-25-06025-f011:**
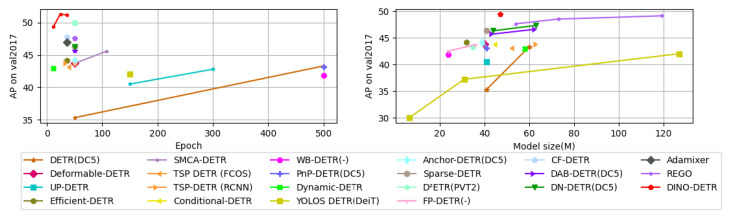
Comparison of all DETR-based detection transformers on the COCO minival set. **left** Performance comparison of detection transformers using a ResNet-50 [80] backbone with regard to training epochs. Networks that are labeled with DC5 take a dilated feature map. **right** Performance comparison of detection transformers with regard to model size (parameters in million).

**Figure 12 sensors-25-06025-f012:**
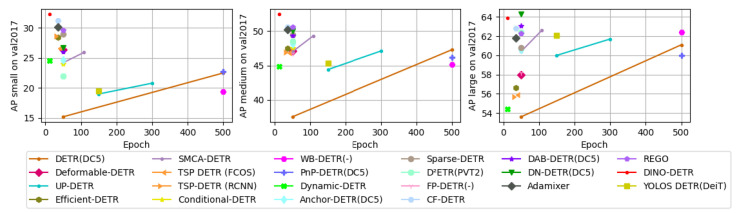
Comparison of DETR-based detection transformers on the COCO minival set using a ResNet-50 backbone. **left** Performance comparison of detection transformers on small objects. **middle** Performance comparison of detection transformers on medium objects. **right** Performance comparison of detection transformers on large objects.

**Table 1 sensors-25-06025-t001:** Overview of improvements in the detection transformer (DETR) to make training convergence faster and improve performance for small objects. Here, Bk represents the backbone, Pre denotes pre-training, Attn indicates attention, and Qry represents the query of the transformer network. Each method represents an improvement over the baseline DETR, and the green check marks indicate where modifications were introduced. The main contributions of each network are summarized in the last column. All GitHub links in this Table are accessed on 25 September 2025.

Methods	Modifications				Publication	Highlights
	Bk	Pre	Attn	Qry		
DETR [11] GitHub https://github.com/facebookresearch/detr	-	-	-	-	ECCV 2020	Transformer, Set-based prediction, bipartite matching
Deformable-DETR [20] GitHub https://github.com/fundamentalvision/Deformable-DETR			✓		ICLR 2021	Deformable-attention module
UP-DETR [21] GitHubhttps://github.com/dddzg/up-detr		✓			CVPR 2021	Unsupervised pre-training, random query patch detection
Efficient-DETR [22]				✓	arXiv 2021	Refence point and top-k queries selection module
SMCA-DETR [23] GitHub https://github.com/gaopengcuhk/SMCA-DETR			✓		ICCV 2021	Spatially-Modulated Co-attention module
TSP-DETR [24] GitHub https://github.com/Edward-Sun/TSP-Detection			✓		ICCV 2021	TSP-FCOS and TSP-RCNN modules for cross attention
Conditional-DETR [25] GitHub https://github.com/Atten4Vis/ConditionalDETR				✓	ICCV 2021	Conditional spatial queries
WB-DETR [26] GitHub https://github.com/aybora/wbdetr	✓				ICCV 2021	Encoder–decoder network without a backbone, LIE-T2T encoder module
PnP-DETR [27] GitHub https://github.com/twangnh/pnp-detr			✓		ICCV 2021	PnP sampling module including pool sampler and poll sampler
Dynamic-DETR [28]			✓		ICCV 2021	Dynamic attention in the encoder–decoder network
YOLOS-DETR [29] GitHub https://github.com/hustvl/YOLOS		✓			NeurIPS 2021	Pre-training encoder network
Anchor-DETR [30] GitHub https://github.com/megvii-research/AnchorDETR			✓	✓	AAAI 2022	Row and Column decoupled-attention, object queries as anchor points
Sparse-DETR [31] GitHub https://github.com/kakaobrain/sparse-detr			✓		ICLR 2022	Cross-attention map predictor, deformable-attention module
$D^{2}$ETR [32] GitHub https://github.com/alibaba/easyrobust/tree/main/ddetr			✓		arXiv 2022	Fine fused features, cross-scale attention module
FP-DETR [33] GitHub https://github.com/encounter1997/FP-DETR	✓	✓			ICLR 2022	Multiscale tokenizer in place of CNN backbone, pre-training encoder network
CF-DETR [34]			✓		AAAI 2022	TEF module to capture spatial relationships, a coarse and a fine layer in the decoder network
DAB-DETR [72] GitHub https://github.com/IDEA-Research/DAB-DETR				✓	ICLR 2022	Dynamic anchor boxes as object queries
DN-DETR [35] GitHub https://github.com/IDEA-Research/DN-DETR				✓	CVPR 2022	Positive noised object queries
AdaMixer [36] GitHub https://github.com/MCG-NJU/AdaMixer			✓		CVPR 2022	3D sampling module, Adaptive mixing module in the decoder
REGO [37] GitHub https://github.com/zhechen/Deformable-DETR-REGO			✓		CVPR 2022	A multi-level recurrent mechanism and a glimpse-based decoder
DINO [38] GitHub https://github.com/facebookresearch/dino				✓	arXiv 2022	Contrastive denoising module, positive and negative noised object queries
Co-DETR [39] GitHub https://github.com/Sense-X/Co-DETR					ICCV 2023	Collaborative hybrid assignments for faster convergence and improved training stability
LW-DETR [40] GitHub https://github.com/Atten4Vis/LW-DETR			✓		arXiv 2024	Lightweight DETR with optimized ViT encoder, shallow decoder, and global attention
RT-DETR [41] GitHub https://github.com/lyuwenyu/RT-DETR			✓	✓	CVPR 2024	Hybrid encoder with multi-scale features, IoU-aware query selection, adaptable inference speed

**Table 2 sensors-25-06025-t002:** Overview of previous surveys on object detection. For each paper, the publication details are provided.

Title	Year	Venue	Description
Advanced Deep-Learning Techniques for Salient and Category-Specific Object Detection: A Survey [50]	2018	SPM	It provides an overview of different object detection domains, including object detection (OD), salient OD, and category-specific OD.
Object Detection in 20 Years: A Survey [73]	2019	TPAMI	This work gives an overview of the evolution of object detectors.
Deep Learning for Generic Object Detection: A Survey [51]	2019	IJCV	A review on deep learning techniques on generic object detection.
A Survey on Deep Learning-based Architectures for Semantic Segmentation on 2D images [53]	2020	PRJ	Deep learning-based methods for semantic segmentation are reviewed.
A Survey of Modern Deep Learning based Object Detection Models [74]	2021	ICV	It briefly overviews deep learning-based (regression-based single-stage and candidate-based two-stage) object detectors.
A Survey of Object Detection Based on CNN and Transformer [70]	2021	PRML	A review of the benefits and drawbacks of deep learning-based object detectors and introduction of transformer-based methods.
Transformers in computational visual media: A survey [71]	2021	CVM	It focuses on backbone design and low-level vision using vision transformer methods.
A survey: object detection methods from CNN to transformer [68]	2022	MTA	Comparison of various CNN-based detection networks and introduction of transformer-based detection networks.
A Survey on Vision Transformer [69]	2023	TPAMI	This paper provides an overview of vision transformers and focuses on summarizing the state-of-the-art research in the field of vision transformers (ViTs).

**Table 3 sensors-25-06025-t003:** Performance comparison of all DETR-based detection transformers on the COCO minival set. Here, networks labeled with DC5 take a dilated feature map. The IoU threshold values are set to 0.5 and 0.75 for AP calculation and also calculate the AP for small ($AP_{s}$), medium ($AP_{m}$), and large ($AP_{l}$) objects. + represents bounding-box refinement and ++ denotes Deformable-DETR. ** indicates Efficient-DETR used 6 encoder layers and 1 decoder layer. S denotes small, and B indicates base. † represents the distillation mechanism by Touvron et al. [227]. ‡ indicates the model is pre-trained on ImageNet-21k. All models use 300 queries, while DETR uses 100 object queries to input to the decoder network. The models with superscript * use three pattern embeddings. All GitHub links in this Table are accessed on 25 September 2025.

Methods	Backbone	Publications	Epoch	GFLOPs	Parameters (M)	AP	AP^50^	AP^75^	AP_S_	AP_M_	AP_L_
	DC5-ResNet-50		50	187	41	35.3	55.7	36.8	15.2	37.5	53.6
DETR [11] GitHub https://github.com/facebookresearch/detr	DC5-ResNet-50	ECCV 2020	500	187	41	43.3	63.1	45.9	22.5	47.3	61.1
	DC5-ResNet-101		500	253	60	44.9	64.7	47.7	23.7	49.5	62.3
	ResNet-50		50	173	40	43.8	62.6	47.7	26.4	47.1	58.0
Deformable-DETR [20] GitHub https://github.com/fundamentalvision/Deformable-DETR	ResNet-50 +	ICLR 2021	50	173	40	45.4	64.7	49.0	26.8	48.3	61.7
	ResNet-50 ++		50	173	40	46.2	65.2	50.0	28.8	49.2	61.7
UP-DETR [21] GitHub https://github.com/dddzg/up-detr	ResNet-50	CVPR 2021	150	86	41	40.5	60.8	42.6	19.0	44.4	60.0
	ResNet-50		300	86	41	42.8	63.0	45.3	20.8	47.1	61.7
	ResNet-50		36	159	32	44.2	62.2	48.0	28.4	47.5	56.6
Efficient-DETR [22]	ResNet-101	arXiv 2021	36	239	51	45.2	63.7	48.8	28.8	49.1	59.0
	ResNet-101 **		36	289	54	45.7	64.1	49.5	28.2	49.1	60.2
	ResNet-50		50	152	40	43.7	63.6	47.2	24.2	47.0	60.4
SMCA-DETR [23] GitHub https://github.com/gaopengcuhk/SMCA-DETR	ResNet-50	ICCV 2021	108	152	40	45.6	65.5	49.1	25.9	49.3	62.6
	ResNet-101		50	218	58	44.4	65.2	48.0	24.3	48.5	61.0
TSP-DETR [24] GitHub https://github.com/Edward-Sun/TSP-Detection	FCOS-ResNet-50	ICCV 2021	36	189	51.5	43.1	62.3	47.0	26.6	46.8	55.9
	RCNN-ResNet-50		36	188	63.6	43.8	63.3	48.3	28.6	46.9	55.7
Conditional-DETR [25] GitHub https://github.com/Atten4Vis/ConditionalDETR	DC5-ResNet-50	ICCV 2021	50	195	44	43.8	64.4	46.7	24.0	47.6	60.7
	DC5-ResNet-101		50	262	63	45.0	65.5	48.4	26.1	48.9	62.8
WB-DETR [26] GitHub https://github.com/aybora/wbdetr	-	ICCV 2021	500	98	24	41.8	63.2	44.8	19.4	45.1	62.4
PnP-DETR [27] GitHub https://github.com/twangnh/pnp-detr	DC5-ResNet-50	ICCV 2021	500	145	41	43.1	63.4	45.3	22.7	46.5	61.1
Dynamc-DETR [28]	ResNet-50	ICCV 2021	12	-	58	42.9	61.0	46.3	24.6	44.9	54.4
YOLOS-DETR [29] GitHub https://github.com/hustvl/YOLOS	DeiT-S [227] †	NeurIPS 2021	150	194	31	36.1	56.5	37.1	15.3	38.5	56.2
	DeiT-B [227] †		150	538	127	42.0	62.2	44.5	19.5	45.3	62.1
Anchor-DETR [30] GitHub https://github.com/megvii-research/AnchorDETR	DC5-ResNet-50 *	AAAI 2022	50	151	39	44.2	64.7	47.5	24.7	48.2	60.6
	DC5-ResNet-101 *		50	237	58	45.1	65.7	48.8	25.8	49.4	61.6
Sparse-DETR [31] GitHub https://github.com/kakaobrain/sparse-detr	ResNet-50-ρ-0.5	ICLR 2022	50	136	41	46.3	66.0	50.1	29.0	49.5	60.8
	Swin-T-ρ-0.5 [228]		50	144	41	49.3	69.5	53.3	32.0	52.7	64.9
$D^{2}$ETR [32] GitHub https://github.com/alibaba/easyrobust/tree/main/ddetr	PVT2	arXiv 2022	50	82	35	43.2	62.9	46.2	22.0	48.5	62.4
Def $D^{2}$ETR [32]	PVT2		50	93	40	50.0	67.9	54.1	31.7	53.4	66.7
FP-DETR-S [33] GitHub https://github.com/encounter1997/FP-DETR	-		50	102	24	42.5	62.6	45.9	25.3	45.5	56.9
FP-DETR-B [33] GitHub https://github.com/encounter1997/FP-DETR	-	ICLR 2022	50	121	36	43.3	63.9	47.7	27.5	46.1	57.0
FP-DETR-B ‡ [33] GitHub https://github.com/encounter1997/FP-DETR	-		50	121	36	43.7	64.1	47.8	26.5	46.7	58.2
CF-DETR [34]	ResNet-50	AAAI 2022	36	-	-	47.8	66.5	52.4	31.2	50.6	62.8
	ResNet-101		36	-	-	49.0	68.1	53.4	31.4	52.2	64.3
DAB-DETR [72] GitHub https://github.com/IDEA-Research/DAB-DETR	DC5-ResNet-50 *	ICLR 2022	50	216	44	45.7	66.2	49.0	26.1	49.4	63.1
	DC5-ResNet-101 *		50	296	63	46.6	67.0	50.2	28.1	50.5	64.1
DN-DETR [35] GitHub https://github.com/IDEA-Research/DN-DETR	ResNet-50	CVPR 2022	50	94	44	44.1	64.4	46.7	22.9	48.0	63.4
	DC5-ResNet-50		50	202	44	46.3	66.4	49.7	26.7	50.0	64.3
	ResNet-101		50	174	63	45.2	65.5	48.3	24.1	49.1	65.1
	DC5-ResNet-101		50	282	63	47.3	67.5	50.8	28.6	51.5	65.0
AdaMixer [36] GitHub https://github.com/MCG-NJU/AdaMixer	ResNet-50	CVPR 2022	36	132	139	47.0	66.0	51.1	30.1	50.2	61.8
	ResNeXt-101-DCN		36	214	160	49.5	68.9	53.9	31.3	52.3	66.3
	Swin-s [228]		36	234	164	51.3	71.2	55.7	34.2	54.6	67.3
REGO [37] GitHub https://github.com/zhechen/Deformable-DETR-REGO	ResNet-50 ++	CVPR 2022	50	190	54	47.6	66.8	51.6	29.6	50.6	62.3
	ResNet-101 ++		50	257	73	48.5	67.0	52.4	29.5	52.0	64.4
	ReNeXt-101 ++		50	434	119	49.1	67.5	53.1	30.0	52.6	65.0
DINO [38] GitHub https://github.com/facebookresearch/dino	ReNet-50-4scale *	arXiv 2022	12	279	47	49.0	66.6	53.5	32.0	52.3	63.0
	ResNet-50-5scale *		12	860	47	49.4	66.9	53.8	32.3	52.5	63.9
	ReNet-50-5scale *		24	860	47	51.3	69.1	56.0	34.5	54.2	65.8
	ResNet-50-5scale *		36	860	47	51.2	69.0	55.8	35.0	54.3	65.3
Co-DETR [39] GitHub https://github.com/Sense-X/Co-DETR	ReNet-50 *	ICCV 2023	12	279	47	52.1	69.3	57.3	35.4	55.5	67.2
	ReNet-50 *		36	860	47	54.8	72.5	60.1	38.3	58.4	69.6
	Swin-L(IN-22K) *		12	860	47	59.3	77.3	64.9	43.3	63.3	75.5
	Swin-L(IN-22K) *		24	860	47	60.4	78.3	66.4	44.6	64.2	76.5
	Swin-L(IN-22K) *		36	860	47	60.7	78.5	66.7	45.1	64.7	76.4
LW-DETR [40] GitHub https://github.com/Atten4Vis/LW-DETR	-	arXiv 2024	50	67.7	54.6	54.4	-	-	48.0	52.5	56.1
RT-DETR [41] GitHub https://github.com/lyuwenyu/RT-DETR	ReNet-50*	CVPR 2024	72	136	42	53.1	71.3	57.7	34.8	58.0	70.0
	ResNet-101 *		72	259	76	54.3	72.7	58.6	36.0	58.8	72.1
RT-DETRv2 [224] GitHub https://github.com/supervisely-ecosystem/RT-DETRv2	ReNet-50 *	arXiv 2024	72	136	42	53.4	-	-	-	-	-
	ResNet-101 *		72	259	76	54.3	-	-	-	-	-
RT-DETRv3 [225] GitHub https://github.com/clxia12/RT-DETRv3	ReNet-50 *	arXiv 2024	72	136	42	53.4	-	-	-	-	-
	ResNet-101 *		72	259	76	54.6	-	-	-	-	-

**Table 4 sensors-25-06025-t004:** Overview of the advantages and limitations of detection transformers. All GitHub links in this Table are accessed on 25 September 2025.

Methods	Publications	Advantages	Limitations
DETR [11] GitHub https://github.com/facebookresearch/detr	ECCV 2020	Removes the need for hand-designed components like NMS or anchor generation.	Low performance on small objects and slow training convergence.
Deformable-DETR [20] GitHub https://github.com/fundamentalvision/Deformable-DETR	ICLR 2021	Deformable attention network, which makes training convergence faster.	Number of encoder tokens increases by 20 times compared to DETR.
UP-DETR [21] GitHub https://github.com/dddzg/up-detr	CVPR 2021	Pre-training for Multi-tasks learning and Multi-queries localization.	Pre-training for patch localization, CNN and transformers pre-training needs to integrate.
Efficient-DETR [22]	arXiv 2021	Reduces decoder layers by employing dense and sparse set based network	Increase in GFLOPs twice compared to original DETR.
SMCA-DETR [23] GitHub https://github.com/gaopengcuhk/SMCA-DETR	ICCV 2021	Regression-aware mechanism to increase convergence speed	Low performance in detecting small objects.
TSP-DETR [24] GitHub https://github.com/Edward-Sun/TSP-Detection	ICCV 2021	Deals with issues of Hungarian loss and the cross-attention mechanism of Transformer.	Uses proposals in TSP-FCOS and feature points in TSP-RCNN as in CNN-based detectors.
Conditional-DETR [25] GitHub https://github.com/Atten4Vis/ConditionalDETR	ICCV 2021	Conditional queries remove dependency on content embeddings and ease the training.	Performs better than DETR and deformable-DETR for stronger backbones.
WB-DETR [26] GitHub https://github.com/aybora/wbdetr	ICCV 2021	Pure transformer network without backbone.	Low performance on small objects.
PnP-DETR [27] GitHub https://github.com/twangnh/pnp-detr	ICCV 2021	Sampling module provides foreground and a small quantity of background features.	Breaks 2d spatial structure by taking foreground tokens and reducing background tokens.
Dynamic-DETR [28]	ICCV 2021	Dynamic attention provides small feature resolution and improves training convergence.	Still dependent on CNN networks as convolution-based encoder and an ROI-based decoder.
YOLOS-DETR [29] GitHub https://github.com/hustvl/YOLOS	NeurIPS 2021	Convert ViT pre-trained on ImageNet-1k dataset into Object detector.	Pre-trained ViT still needs improvements as it requires long training epochs.
Anchor-DETR [30] GitHub https://github.com/megvii-research/AnchorDETR	AAAI 2022	Object queries as anchor points that predict multiple objects at one position.	Consider queries as 2D anchor points which ignore object scale.
Spare-DETR [31] GitHub https://github.com/kakaobrain/sparse-detr	ICLR 2022	Improve performance by updating tokens referenced by the decoder.	Performance is strongly dependent on the backbone specifically for large objects.
$D^{2}$ETR [32] GitHub https://github.com/alibaba/easyrobust/tree/main/ddetr	arXiv 2022	Decoder-only transformer network to reduce computational cost.	Decreases computation comlexity significantly but has low performance on small objects.
FP-DETR [33] GitHub https://github.com/encounter1997/FP-DETR	ICLR 2022	Pre-Training of the encoder-only transformer.	Low performance on large objects.
CF-DETR [34] GitHub https://github.com/facebookresearch/detr	AAAI 2022	Refine coarse features to improve localization accuracy of small objects.	Addition of three new modules increase network size.
DAB-DETR [72] GitHub https://github.com/IDEA-Research/DAB-DETR	ICLR 2022	Anchor-boxes as queries, attention for different scale objects.	Positional prior for only foreground objects.
DN-DETR [35] GitHub https://github.com/IDEA-Research/DN-DETR	CVPR 2022	Denoising training for positional-prior for foreground and background regions.	Denoising training by adding positive noise to object queries ignoring background regions.
AdaMixer [36] GitHub https://github.com/MCG-NJU/AdaMixer	CVPR 2022	Faster Convergence, Improves the adaptability of query-based decoding mechanism.	Large number of parameters.
REGO [37] GitHub https://github.com/zhechen/Deformable-DETR-REGO	CVPR 2022	Attention mechanism gradually focus on foreground regions more accurately.	Multi-stage RoI-based attention modeling increases the number of parameters.
DINO [38] GitHub https://github.com/facebookresearch/dino	arXiv 2022	impressive results on small and medium-sized datasets	Performance drops for large size objects
Co-DETR [39] GitHub https://github.com/Sense-X/Co-DETR	ICCV 2023	Enhances encoder feature learning and decoder attention via collaborative hybrid assignments.	Increases training complexity due to multiple assignment heads.
LW-DETR [40] GitHub https://github.com/Atten4Vis/LW-DETR	arXiv 2024	Achieves real-time detection with a lightweight transformer design using optimized ViT encoder and window attention.	Limited evaluation on benchmarks; less mature than YOLO-style detectors.

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
