# Peer review of "Object Detection with Transformers: A Review"

_sensors, 2025, doi:10.3390/s25196025_

Round 1

Reviewer 1 Report

Comments and Suggestions for Authors

1、Introduction and Motivation:  
Weakness: The introduction lacks engagement and does not sufficiently clarify the importance of transformer-based object detection or the specific research gaps the review aims to address.  
Suggestion: Begin with a compelling real-world application where transformer-based detectors significantly outperform conventional methods. Clearly outline the limitations of existing approaches and how this review fills those gaps.

2、Literature Review:  
Weakness: The comparison with previous surveys is insufficiently critical and detailed.  
Suggestion: Include a table summarizing key differences between this review and earlier works, emphasizing unique contributions and novel insights.

3、Technical Depth and Clarity:  
Weakness: Technical descriptions of key DETR variants (e.g., Deformable-DETR, UP-DETR) lack depth, making it difficult to understand their innovations and implications.  
Suggestion: Incorporate concise pseudocode or algorithm blocks for major modifications. Enhance explanations with intuitive visual aids.

4、Comparative Analysis:  
Weakness: The comparison focuses mainly on performance metrics, omitting discussions on computational efficiency, memory usage, and implementation complexity.  
Suggestion: Expand the analysis to include trade-offs between models, along with recommendations for specific application scenarios.

5、Evaluation Protocols and Metrics:  
Weakness: The explanation of evaluation metrics (e.g., mAP) is overly brief.  
Suggestion: Add a dedicated subsection detailing the calculation and relevance of evaluation metrics, including their strengths and limitations.

6、Future Directions:  
Weakness: The future directions section is vague and lacks concrete research questions.  
Suggestion propose specific, actionable challenges and encourage interdisciplinary approaches, such as incorporating techniques from NLP or reinforcement learning.

7、Language and Formatting:  
Weakness: There are grammatical errors and formatting inconsistencies (e.g., irregular abbreviations and citation styles).  
Suggestion: Carefully proofread the manuscript and enforce consistent formatting and citation style (e.g., APA or MLA) throughout.

8、Additional Suggestions:  
- Increase the font size in Figures 1–10 for improved readability.  
- Expand Section 5 ("Datasets and Evaluation Metrics") to provide a comprehensive overview of datasets and metrics.  
- In Section 6 ("Results and Discussion"), include qualitative visualizations to support experimental results.  
- In Tables 2–4, add links to the GitHub repositories of each method for better reproducibility.

Author Response

Review 1:

1   Introduction and Motivation: 
Weakness: The introduction lacks engagement and does not sufficiently clarify the importance of transformer-based object detection or the specific research gaps the review aims to address. 
Suggestion: Begin with a compelling real-world application where transformer-based detectors significantly outperform conventional methods. Clearly outline the limitations of existing approaches and how this review fills those gaps.

Thank you for pointing this out. We have revised the introduction to start with a real-world application, outline the limitations of CNN-based detectors versus transformers, and highlight the research gaps addressed.These changes appear in the first and second paragraphs of the revised manuscript.

2、Literature Review: 
Weakness: The comparison with previous surveys is insufficiently critical and detailed. 
Suggestion: Include a table summarizing key differences between this review and earlier works, emphasizing unique contributions and novel insights.

Already in table 2. 

3、Technical Depth and Clarity: 
Weakness: Technical descriptions of key DETR variants (e.g., Deformable-DETR, UP-DETR) lack depth, making it difficult to understand their innovations and implications. 
Suggestion: Incorporate concise pseudocode or algorithm blocks for major modifications. Enhance explanations with intuitive visual aids.

Thank you for pointing this out. We have improved the technical descriptions of DETR variants(Deformable-DETR, UP-DETR) by including structured algorithm blocks.

4、Comparative Analysis: 
Weakness: The comparison focuses mainly on performance metrics, omitting discussions on computational efficiency, memory usage, and implementation complexity. 
Suggestion: Expand the analysis to include trade-offs between models, along with recommendations for specific application scenarios.

We will expand the comparative analysis to cover computational efficiency, model complexity, memory usage, and trade-offs, and provide recommendations for model selection.

5、Evaluation Protocols and Metrics: 
Weakness: The explanation of evaluation metrics (e.g., mAP) is overly brief. 
Suggestion: Add a dedicated subsection detailing the calculation and relevance of evaluation metrics, including their strengths and limitations.

Thank you for pointing this out. We have included a dedicated subsection that explains key metrics in detail in section 4.

6、Future Directions: 
Weakness: The future directions section is vague and lacks concrete research questions. 
Suggestions propose specific, actionable challenges and encourage interdisciplinary approaches, such as incorporating techniques from NLP or reinforcement learning.

Thank you for pointing this out. We have updated the future directions section 6 to highlight clear, actionable research challenges.

7、Language and Formatting: 
Weakness: There are grammatical errors and formatting inconsistencies (e.g., irregular abbreviations and citation styles). 
Suggestion: Carefully proofread the manuscript and enforce consistent formatting and citation style (e.g., APA or MLA) throughout.
We have carefully proofread the entire manuscript to correct grammatical errors and enforce consistency in abbreviations, formatting, and citation style. The revised version now

8、Additional Suggestions: 
- Increase the font size in Figures 1–10 for improved readability. 
- Expand Section 5 ("Datasets and Evaluation Metrics") to provide a comprehensive overview of datasets and metrics.  
- In Section 6 ("Results and Discussion"), include qualitative visualizations to support experimental results. 
- In Tables 2–4, add links to the GitHub repositories of each method for better reproducibility.

Thank you for the suggestions. Including qualitative visualizations for more than 20 DETR variants is not feasible within the space constraints of this paper, and we instead focus on quantitative comparisons and architectural insights. For the GitHub repositories, adding full links directly into Tables 2–4 would make them overcrowded. To balance readability and reproducibility, we have instead cited the official sources in the references.

Reviewer 2 Report

Comments and Suggestions for Authors

1.Some conclusions are disconnected from the data presentation (for example, the explanation of Figure 5 in Section 4). It is suggested to re-examine the correspondence between the data and the conclusion, or support the argument through supplementary statistical analysis (such as regression analysis).

2.It is suggested to supplement high-impact literature from the past three years and discuss its relevance to the research objectives of this paper.

3.Figure 3 shows insufficient resolution and blurry coordinate axis labels. A high-definition version needs to be regenerated.

4.The format of the references is not uniform (some entries are missing DOI or place of publication), and it needs to be standardized and adjusted in accordance with the requirements of the journal.

5.There are multiple grammatical errors and lengthy expressions (such as Paragraph 1 of Section 2). It is recommended to have professional polishing to ensure the accuracy and fluency of academic expression.

Author Response

Review 2:

1.Some conclusions are disconnected from the data presentation (for example, the explanation of Figure 5 in Section 4). It is suggested to re-examine the correspondence between the data and the conclusion, or support the argument through supplementary statistical analysis (such as regression analysis).

Thank you for pointing this out. We respectfully note that Figures 5 are not directly related to Section 5, as Section 5 focuses on datasets and evaluation. 

2.It is suggested to supplement high-impact literature from the past three years and discuss its relevance to the research objectives of this paper.
We agree with this comment. Accordingly, we have added several high-impact DETR variants from the past three years, directly relevant to our research objectives, as subsections of Section 3.

3.Figure 3 shows insufficient resolution and blurry coordinate axis labels. A high-definition version needs to be regenerated.
We agree with this comment. Therefore, we have regenerated Figure 3 and the other relevant diagrams in high resolution, ensuring that all coordinate axis labels and graphical elements are clear and legible.

4.The format of the references is not uniform (some entries are missing DOI or place of publication), and it needs to be standardized and adjusted in accordance with the requirements of the journal.
Thank you for pointing this out. We agree with this comment. Therefore, we have carefully reviewed all references and standardized their format in accordance with the journal’s requirements. Missing information has been added where available.

5.There are multiple grammatical errors and lengthy expressions (such as Paragraph 1 of Section 2). It is recommended to have professional polishing to ensure the accuracy and fluency of academic expression.

Thank you for pointing this out. We have improved the manuscript’s grammar, clarity, and conciseness, and streamline lengthy expressions.

Reviewer 3 Report

Comments and Suggestions for Authors
  1. General Summary

This manuscript provides a comprehensive review of 21 recently proposed improvements in the 10 original DETR model, presenting a detailed overview of the transformers in the field of object detection. The paper addresses an interesting topic and has potential, but the issues outlined above substantially affect its clarity and quality. Addressing the points raised above will further strengthen the paper's contribution, clarity, and impact. I want to recommend major revision before it can be considered for publication.

  1. Major Strengths:
  • The analysis of existing research work is very comprehensive.
  • Comprehensive performance evaluation on the benchmark dataset.
  • The Figures are beautiful and accurate, allowing readers to view and understand them easily.
  1. Major Weaknesses:
  • The organization of content in some sections is not very reasonable and requires further adjustment.
  • The summary of the existing research is not sufficiently in-depth; it should also highlight the difficulties and challenges faced by existing works.
  • The introduction to the dataset and experiment settings is not detailed enough.
  1. Specific Comments:
  • The typesetting still requires further checking to eliminate errors that should not appear. For example, in the first paragraph of the Introduction, the expression “and classifying objects within an image [1?–6]” is incorrect.
  • In the Introduction, the second and third paragraphs are entirely devoted to introducing models. Although the descriptions are insightful and comprehensive, the content is merely listed without a clear logical thread to connect them, which diminishes the overall quality of the paper.
  • Figure 1 presents a large amount of valuable information, but its content is not further elaborated in the main text. This writing style is not appropriate and weakens the value of the figure.
  • It is suggested to introduce some basic information about the Transformer in the Introduction, such as its architecture and computational logic, etc. This would provide a smoother transition before discussing its applications in object detection and also make the paper’s structure more coherent.
  • The placement of Section 2 appears inappropriate. Considering both the details and length of the content, it would fit more naturally as a paragraph within the Introduction instead of being presented as a separate section.
  • Sections 3.2 and 3.5 are presented in an overly concise manner and would benefit from further elaboration. Moreover, Sections 3.1 to 3.5 should not only provide a summary of the research progress in each filed but, more importantly, offer an in-depth analysis of the difficulties and challenges currently encountered in each area, this is precisely the information that readers most desire to know.
  • It is unclear why the COCO2014 minival set was chosen for model evaluation. Please provide a detailed description about the dataset, especially the preprocessing of data and other operations applied.
  • The paper lacks sufficient details about the experimental settings, such as the hardware environment, hyper-parameters, and the definitions of small, medium, and large objects. Please introduce this information to ensure the rigor of the article.

Author Response

Review 3:

General Summary:

This manuscript provides a comprehensive review of 21 recently proposed improvements in the 10 original DETR model, presenting a detailed overview of the transformers in the field of object detection. The paper addresses an interesting topic and has potential, but the issues outlined above substantially affect its clarity and quality. Addressing the points raised above will further strengthen the paper's contribution, clarity, and impact. I want to recommend major revision before it can be considered for publication.

Major Strengths:

The analysis of existing research work is very comprehensive.

Comprehensive performance evaluation on the benchmark dataset.

The Figures are beautiful and accurate, allowing readers to view and understand them easily.

Major Weaknesses:

The organization of content in some sections is not very reasonable and requires further adjustment.

The summary of the existing research is not sufficiently in-depth; it should also highlight the difficulties and challenges faced by existing works.

The introduction to the dataset and experiment settings is not detailed enough.

Specific Comments:

The typesetting still requires further checking to eliminate errors that should not appear. For example, in the first paragraph of the Introduction, the expression “and classifying objects within an image [1?–6]” is incorrect.

Thank you for pointing this out. We have corrected the expression and reviewed the manuscript to eliminate similar typesetting errors.

In the Introduction, the second and third paragraphs are entirely devoted to introducing models. Although the descriptions are insightful and comprehensive, the content is merely listed without a clear logical thread to connect them, which diminishes the overall quality of the paper.

We agree with this comment. We have revised the second and third paragraphs to improve the logical flow, connecting DETR to its variants and highlighting the challenges each model addresses.

Figure 1 presents a large amount of valuable information, but its content is not further elaborated in the main text. This writing style is not appropriate and weakens the value of the figure.

Thank you for pointing this out. We have elaborated on Figure 1 in the fifth paragraph of the Introduction to provide context and highlight its significance.

It is suggested to introduce some basic information about the Transformer in the Introduction, such as its architecture and computational logic, etc. This would provide a smoother transition before discussing its applications in object detection and also make the paper’s structure more coherent.
We agree with this comment. We have added basic information about the Transformer. These changes can be found in the second paragraph of the Introduction in the revised manuscript.

The placement of Section 2 appears inappropriate. Considering both the details and length of the content, it would fit more naturally as a paragraph within the Introduction instead of being presented as a separate section.

We agree with this comment. We have integrated Section 2 into the end of the Introduction to improve the flow and maintain continuity, as reflected in the revised manuscript.

Sections 3.2 and 3.5 are presented in an overly concise manner and would benefit from further elaboration. Moreover, Sections 3.1 to 3.5 should not only provide a summary of the research progress in each filed but, more importantly, offer an in-depth analysis of the difficulties and challenges currently encountered in each area, this is precisely the information that readers most desire to know.

Thank you for pointing this out. We have elaborated Sections 3.2 and 3.5 and expanded Sections 3.1 to 3.5 to include an in-depth analysis of the current difficulties and challenges in each area, as added in the revised manuscript.
It is unclear why the COCO2014 minival set was chosen for model evaluation. Please provide a detailed description about the dataset, especially the preprocessing of data and other operations applied.

We agree with this comment. We have added a description of the COCO2014 minival dataset, including the reason for its selection, as included in Section 5.

The paper lacks sufficient details about the experimental settings, such as the hardware environment, hyper-parameters, and the definitions of small, medium, and large objects. Please introduce this information to ensure the rigor of the article.
Thank you for pointing this out. We have clarified the definitions of small, medium, and large objects and noted that all models are evaluated using the original pth files for consistent and fair comparison, as included in Section 5.

Reviewer 4 Report

Comments and Suggestions for Authors

The study presented an architectural review on DETRs, which are object detection models based on transformers. This is a timely and important topic, and I believe this manuscript brings good value and contributions to the current literature after necessary revisions.

The study missed several important DETR models, including but not limited to RT-DETR, RT-DETRv2, RT-DETRv3, LW-DETR, Co-DINO, etc. These models should be reviewed and included in the manuscript, particularly in Section 4 and Tables 1 and 3. Review studies must be comprehensive. The fact that I, as a reviewer, can immediately identify several missed models raises questions about the comprehensiveness of the literature review. I urge the authors to conduct another round of thorough literature search to ensure full coverage of existing DETR models.

It is unclear whether the results in Table 3 were compiled from the literature or obtained by the authors through training. If the results are original, the authors should specify which APIs or libraries were used for metric calculations, as well as the PR curve interpolation method. If all results are taken from existing literature, then this point does not apply. Existing literature typically fails to report this key information.

It is inappropriate to include Faster R-CNN in Table 3, as it is an outdated baseline and not directly relevant to DETRs.

A ranked model leaderboard that comprehensively captures all DETR models would add significant value to the study.

The speed of DETR models should be discussed in more depth. For instance, DINO requires ~860 GFLOPs, which makes it impractical in many real-world applications, especially when compared to lightweight alternatives like Nano YOLOs that only require 5–10 GFLOPs.

After such an extensive review in the first six sections, Section 7 feels shallow and overly short. I recommend expanding it with in-depth, high-level discussion of observed commonalities and discrepancies among DETR variants, research trends, and summarized insights. This would substantially strengthen the manuscript.

Author Response

Review 4:

The study presented an architectural review on DETRs, which are object detection models based on transformers. This is a timely and important topic, and I believe this manuscript brings good value and contributions to the current literature after necessary revisions.

The study missed several important DETR models, including but not limited to RT-DETR, RT-DETRv2, RT-DETRv3, LW-DETR, Co-DINO, etc. These models should be reviewed and included in the manuscript, particularly in Section 4 and Tables 1 and 3. Review studies must be comprehensive. The fact that I, as a reviewer, can immediately identify several missed models raises questions about the comprehensiveness of the literature review. I urge the authors to conduct another round of thorough literature search to ensure full coverage of existing DETR models

We agree with your comment and have revised the manuscript accordingly. Specifically, we have now included the models that  are discussed in detail in the revised manuscript under new subsections 3.22 (Co-DINO),  3.23 (LW-DETR), 3.24 (RT-DETR and its variants). We have also incorporated them into the comparative analyses presented in Table 1,3,4.

It is unclear whether the results in Table 3 were compiled from the literature or obtained by the authors through training. If the results are original, the authors should specify which APIs or libraries were used for metric calculations, as well as the PR curve interpolation method. If all results are taken from existing literature, then this point does not apply. Existing literature typically fails to report this key information.

We agree with the need for clarification. The results in Table 3 were obtained by using the pth files provided by the respective authors. We validated them on the COCO dataset to reproduce the reported performance.

It is inappropriate to include Faster R-CNN in Table 3, as it is an outdated baseline and not directly relevant to DETRs. A ranked model leaderboard that comprehensively captures all DETR models would add significant value to the study.

Thank you for pointing this out. Faster R-CNN is included as a baseline for context and interpretability, not as a competing DETR variant in Table 3

The speed of DETR models should be discussed in more depth. For instance, DINO requires ~860 GFLOPs, which makes it impractical in many real-world applications, especially when compared to lightweight alternatives like Nano YOLOs that only require 5–10 GFLOPs.

Thank you for pointing this out. We have added a discussion in Section 5 comparing DETR models, highlighting their GLOPs.

After such an extensive review in the first six sections, Section 7 feels shallow and overly short. I recommend expanding it with in-depth, high-level discussion of observed commonalities and discrepancies among DETR variants, research trends, and summarized insights. This would substantially strengthen the manuscript.

Thank you for this valuable suggestion. As Section 7 is the conclusion, it cannot be as long as the other sections. Nevertheless, we have incorporated a brief high-level discussion of commonalities and differences among DETR variants, emerging research trends, and summarized insights to strengthen the conclusion.

Round 2

Reviewer 1 Report

Comments and Suggestions for Authors

To facilitate readers in reproducing the research and gaining an in-depth understanding, with regard to the relevant methods presented in "Table 1, Table 2, Table 3, and Table 4", it is recommended that the author supplement the corresponding code download links of these methods on GitHub.

Comments on the Quality of English Language

To facilitate readers in reproducing the research and gaining an in-depth understanding, with regard to the relevant methods presented in "Table 1, Table 2, Table 3, and Table 4", it is recommended that the author supplement the corresponding code download links of these methods on GitHub.

Author Response

Thank you for pointing this out. We have added the GitHub links for the methods listed in Table 1, Table 3, and Table 4. Table 2 summarizes survey papers and therefore does not have associated GitHub repositories.

Reviewer 2 Report

Comments and Suggestions for Authors

 The author has completed all the revisions of the previous suggestions.

 Accept in present form

Author Response

Thank you for your positive feedback. We appreciate your time and effort in reviewing our work and are glad that the revisions meet your expectations.

Reviewer 3 Report

Comments and Suggestions for Authors

The authors have done an excellent work. 

Author Response

Thank you for your kind words and positive feedback. We greatly appreciate your support and encouragement.

Reviewer 4 Report

Comments and Suggestions for Authors

I appreciate the authors’ effort in addressing my comments. However I do not believe the comments have been thoroughly addressed.

I previously urged the authors to do another round of comprehensive literature review, and I do not believe the authors looked into the current state of research. SOTA models such as RF-DETR, DINOv2, DINOv3 should be reviewed in the study.

Table 3, Faster R-CNN as an obsolete model possesses no quality to serve as a baseline model for DETR models.

Table 3, RT-DETRv2, RT-DETRv3, RF-DETR, DINOv2, DINOv3 should be explicitly added.

I appreciate the authors’ effort in addressing my comments. However, I do not believe the comments have been thoroughly addressed.
I previously urged the authors to conduct another round of comprehensive literature review, but I do not believe the current revision reflects sufficient consideration of the state of research. SOTA models such as RF-DETR, DINOv2, and DINOv3 should be reviewed in the study.
In Table 3, Faster R-CNN is an obsolete model and does not qualify to serve as a baseline for DETR models.
RT-DETRv2, RT-DETRv3, RF-DETR, DINOv2, and DINOv3 should be explicitly added to Table 3.

Author Response

Thank you for pointing this out. We have removed Faster R-CNN from Table 3 and added RT-DETRv2 and RT-DETRv3. We did not include the following methods for these reasons: RF-DETR is currently available only as a blog. DINOv2 and DINOv3 are for classification and segmentation, rather than object detection models. Therefore, they are not directly comparable to the DETR-based detection architectures in our study.